# WHEN DOES MULTIMODALITY LEAD TO BETTER TIME SERIES FORECASTING?

## ABSTRACT

Recently, there has been growing interest in incorporating textual information into foundation models for time series forecasting. However, it remains unclear whether and under what conditions such multimodal integration consistently yields gains. We systematically investigate these questions across a diverse benchmark of 16 forecasting tasks spanning 7 domains, including health, environment, and economics. We evaluate two popular multimodal forecasting paradigms: aligning-based methods, which align time series and text representations; and prompting-based methods, which directly prompt large language models for forecasting. Our findings reveal that the benefits of multimodality are highly condition-dependent. While we confirm reported gains in some settings, these improvements are not universal across datasets or models. To move beyond empirical observations, we disentangle the effects of model architectural properties and data characteristics, drawing data-agnostic insights that generalize across domains. Our findings highlight that on the modeling side, incorporating text information is most helpful given (1) high-capacity text models, (2) comparatively weaker time series models, and (3) appropriate aligning strategies. On the data side, performance gains are more likely when (4) sufficient training data is available and (5) the text offers complementary predictive signal beyond what is already captured from the time series alone. Our study offers a rigorous, quantitative foundation for understanding when multimodality can be expected to aid forecasting tasks, and reveals that its benefits are neither universal nor always aligned with intuition.

## 1 INTRODUCTION

Multimodal time series (MMTS) forecasting seeks to improve predictive accuracy by combining time series data with auxiliary textual sources, such as clinical notes (Kim et al., 2024), product descriptions (Skenderi et al., 2024), or weather reports (Liu et al., 2024b). Recent studies (Liu et al., 2024b; Cao et al., 2023; Wang et al., 2024; Williams et al., 2025; Zhang et al., 2024b;a) have proposed various MMTS methods, motivated by the intuition that contextual text can enhance the predictive signal beyond what is captured in the time series alone. Existing MMTS approaches fall into two broad categories: aligning-based methods and prompting-based methods. *Aligning-based* methods jointly encode time series and associated text by aligning their representations (Jin et al., 2023; Liu et al., 2024c; Jia et al., 2024). In contrast, *prompting-based* methods directly leverage pre-trained large language models (LLMs) for forecasting by presenting time series and textual descriptions in natural language format (Williams et al., 2025; Xue & Salim, 2023; Requeima et al., 2024).

Despite growing interest in such an area (Liu et al., 2024b; Cao et al., 2023; Wang et al., 2024; Williams et al., 2025; Zhang et al., 2024b; Liu et al., 2025), there remains a lack of systematic understanding of whether, and under what conditions, these approaches truly enhance forecasting performance. Our work is the first of its kind to rigorously evaluate the effectiveness of MMTS across diverse forecasting scenarios. Specifically, we ask (**RQ1**):

*When and how does multimodality really improve forecasting performance?*

To answer this question, we conduct a comprehensive benchmark study on the above two categories of MMTS methods across 16 multimodal datasets that span diverse domains, e.g., health, environment, energy, and economics. In addition, we provide controlled experiments that isolate dataset-level and model-level factors, and derive insights that generalize beyond any single dataset. From the *modeling*

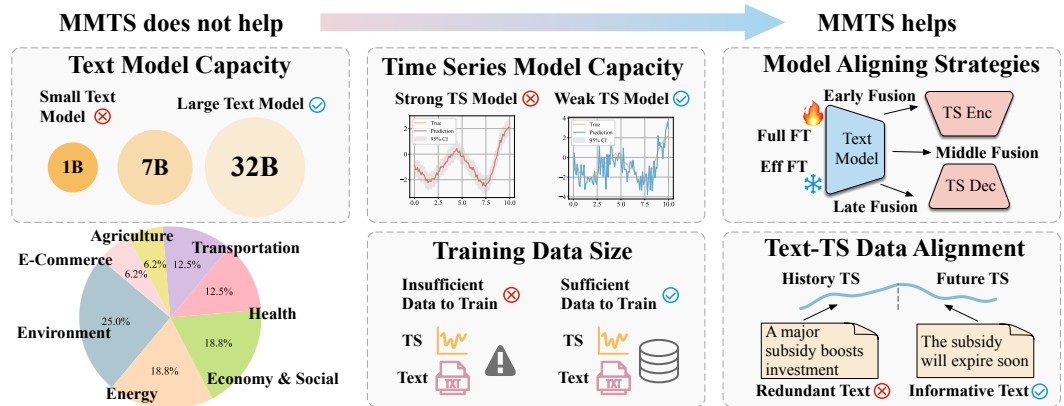

Figure 1: We systematically study when MMTS is effective from modeling and data perspectives, across 16 real-world datasets covering 7 domains. Our findings highlight that MMTS is effective given (1) larger text models, (2) weaker time series models, (3) appropriate aligning strategies, (4) sufficient training data, and (5) complimentary text information.

perspective, we explore: **(RQ2)** How does the capacity of the text encoder affect performance? **(RQ3)** Does the capacity of the time series model matter? **(RQ4)** How important is the design of aligning mechanisms? On the *data* side, we explore: **(RQ5)** How do context length and training sample size influence performance? (**RQ6**) How does text and time series alignment quality affect performance? Figure 1 summarizes our findings. Our findings to the aforementioned questions either provide rigorous, quantitative validation of community assumptions or reveal novel and less intuitive patterns that sharpen the community's understanding of multimodal forecasting. Both aspects facilitate the future development of more effective MMTS models. Our main contributions are the following:

- **Benchmarking and Rethinking MMTS.** We conduct the first large-scale, systematic benchmarking of MMTS forecasting, covering 16 datasets with two major modeling paradigms. Our findings challenge the assumption that multimodal information inherently improves forecast accuracy, showing instead that current benefits are highly context-dependent.
- **Modeling Insights and Scalability Analysis.** We analyze the impact of key modeling factors, including the capacity of text and time series encoders and the choice of fusion strategy.
- **Data Insights.** We identify key data conditions under which textual information enhances forecasting performance. Through data-agnostic controlled experiments, we derive insights that generalize beyond individual benchmarks. Overall, our findings provide practical guidelines on when and how to incorporate text in time series forecasting which motivate and facilitate future research on more effective multimodal approaches.

## 2 RELATED WORKS

MMTS forecasting aims to integrate textual context with time series signals to improve predictive performance. Recent work generally falls into two major modeling paradigms: *aligning-based methods* and *prompting-based methods*. Figure 2 illustrates the differences between unimodal forecasting, alignment-based methods, and prompting-based methods.

**Aligning-Based Methods.** Aligning-based approaches fuse time series and textual information through shared or coordinated latent spaces. GPT4MTS (Jia et al., 2024) incorporates BERT (Devlin et al., 2019b) text embeddings as trainable soft prompts fused with temporally patched input sequences. Time-LLM (Jin et al., 2023) reprograms LLMs by aligning time series embeddings with textual representations. MM-TSFlib (Liu et al., 2024b) explores combinations of different text and time series encoders within a unified benchmark. TimeCMA (Liu et al., 2024a) aligns disentangled time series embeddings with robust prompt-based embeddings. LeRet (Huang et al., 2024) introduces a language-empowered retentive network that models causal dependence. Other methods propose different aligning mechanisms from autoregressive token generation (Liu et al.,

Figure 2: Comparison of unimodal (left) and two MMTS methods (middle, right): aligning-based and prompting-based methods. Aligning-based MMTS aligns time series and text representations; and prompting-based MMTS directly prompts LLMs for forecasting.

2024d), to contrastive objectives (Sun et al., 2023), semantic anchors (Pan et al., 2024) and joint sequence-text predictions (Kim et al., 2024). This paradigm offers architectural flexibility, allowing for tailored encoder designs that can capture complex modality-specific patterns and learn rich joint representations. However, such flexibility comes at the cost of increased design complexity, and effective fusion often requires careful co-design of encoders and alignment mechanisms.

**Prompting-Based Methods.** Prompting-based methods take a different approach by leveraging pre-trained LLMs in a zero- or in-context learning setting. They format both time series and textual input as natural language prompts and rely on the LLM's reasoning capabilities to generate predictions. PromptCast (Xue & Salim, 2023) and LLMTime (Gruver et al., 2024) show that LLMs can be used to produce numerical output for time-series forecasting using carefully designed prompts. LLM Processes (Requeima et al., 2024) elicits numerical predictive distributions from LLMs going beyond one-dimensional time series forecasting to multi-dimensional regression and density estimation. Context is Key (Williams et al., 2025) shows that simply direct prompting the LLM performs better than LLM processes (Requeima et al., 2024) on paired numerical data with diverse types of carefully crafted textual context that contain predictive signals. Prompting-based approaches leverage the generalization capabilities and vast pre-trained knowledge of LLMs, enabling rapid adaptation to new tasks in a zero- or few-shot manner. However, while LLMs are powerful in understanding natural language, they often struggle with numerical reasoning and precise temporal extrapolation.

## 3 MMTS FORMULATION AND METHODS

We formally define the MMTS forecasting task as predicting the future values of a time series conditioned on both historical observations and auxiliary textual information.

- **Time Series.** Let $\mathbf{z}_{1:T} \in \mathbb{R}^{T \times d}$ denote the observed multivariate time series of length $T$ with $d$ variables, and $\mathbf{z}_{T+1:T+\tau} \in \mathbb{R}^{\tau \times d}$ denote the future values to forecast over a horizon $\tau$.
- **Text.** Let $\mathbf{x} \in \mathcal{X}$ denote an associated piece of unstructured textual information (e.g., clinical note, product description, or weather report).
- **MMTS Forecaster.** The goal is to learn a MMTS forecasting function $f$:

$$p(\mathbf{z}_{T+1:T+\tau} \mid \mathbf{z}_{1:T}, \mathbf{x}) = f(\mathbf{z}_{1:T}, \mathbf{x}), f : (\mathbb{R}^{T \times d}, \mathcal{X}) \to \mathcal{P}(\mathbb{R}^{\tau \times d}),$$

that produces a joint distribution over future time series values.

**Aligning-Based Methods.** In the aligning-based paradigm, the function $f$ is composed of separate time series and text encoders, as well as a fusion mechanism. Specifically, these methods define a **time series encoder** $\phi_z : \mathbb{R}^{T \times d} \to \mathbb{R}^h$ that maps the historical time series to a latent representation, a **text encoder** $\phi_x : \mathcal{X} \to \mathbb{R}^h$ that encodes the text into the same latent space, a **fusion mechanism** $\psi : \mathbb{R}^h \times \mathbb{R}^h \to \mathbb{R}^h$ that combines the two modalities (e.g., via addition or concatenation), and a **forecaster** $g : \mathbb{R}^h \to \mathbb{R}^{\tau \times d}$ that maps the fused representation to future predictions:

$$p(\mathbf{z}_{T+1:T+\tau} \mid \mathbf{z}_{1:T}, \mathbf{x}) = g\left(\psi\left(\phi_z(\mathbf{z}_{1:T}), \phi_x(\mathbf{x})\right)\right).$$

**Prompting-Based Methods.** With prompting-based methods, we formally define a **formatter** $\pi : \mathbb{R}^{T \times d} \times \mathcal{X} \to \mathcal{T}$ that converts time series $\mathbf{z}_{1:T}$ and text $\mathbf{x}$ into a textual prompt $\mathbf{t} \in \mathcal{T}$, and we

leverage an **LLM** $l : \mathcal{T} \rightarrow \mathbb{R}^{\tau \times d}$ that generates forecast values (either directly as tokens or indirectly via extraction or parsing of outputs):

$$p(\mathbf{z}_{T+1:T+\tau} \mid \mathbf{z}_{1:T}, \mathbf{x}) = l(\pi(\mathbf{z}_{1:T}, \mathbf{x})).$$

## 4 EMPIRICAL RESULTS

**Players.** For *aligning-based* methods, we explore aligning various unimodal time series models such as PatchTST (Nie et al., 2022), DLinear (Zeng et al., 2023), FEDformer (Zhou et al., 2022), Informer (Zhou et al., 2021), iTransformer (Liu et al., 2023), Chronos (Ansari et al., 2024), with different text models, e.g., BERT (Devlin et al., 2019a), GPT-2 (Radford et al., 2019), T5 (Raffel et al., 2020), Qwen-1.5b (Yang et al., 2024), LLaMA-7b (Touvron et al., 2023). We also compare multimodal models such as Time-LLM (Jin et al., 2023), TimeCMA (Liu et al., 2024a) and LeRet (Huang et al., 2024). For *prompting-based* approaches, we consider directly prompting a wide range of LLMs (LLaMA (Grattafiori & et al, 2024), Qwen (Qwen et al., 2025), Mistral (Jiang et al., 2023), GPT-4o-mini (OpenAI et al., 2024), Claude [1]) with serialized time series and accompanying text. We optimize the prompt following CiK (Williams et al., 2025) (Appendix C.3), and additionally compare in-context learning and chain-of-thought approaches in Table 17. More implementation details can be found at Appendix B (aligning-based approaches) and Appendix C (prompting-based approaches).

**Datasets.** Our benchmark covers 16 real-world datasets spanning diverse domains for both dynamic and static text settings. They also exhibit different context lengths, training data sizes and temporal granularities. Their detailed statistics can be found in Table 4 in Appendix A.

**Evaluation Metrics.** We report Mean Squared Error (MSE) and Mean Absolute Error (MAE) for point forecasts. For models capable of probabilistic forecasting, we additionally report Weighted Quantile Loss (WQL) (Ansari et al., 2024) and Continuous Ranked Probability Score (CRPS) (Williams et al., 2025; Gneiting & Raftery, 2007).

**Overview.** We begin by addressing the central question: *Are MMTS models consistently effective?* Contrary to common expectations or beliefs in the community, we find that multimodal integration does *not* universally lead to improved performance. Beyond an overview of the current landscape, the following sections examine architectural design, alignment strategies, and dataset characteristics. Our findings, either confirming community assumptions or revealing less obvious patterns, yield insights that generalize beyond current datasets on when multimodality aids forecasting.

### 4.1 ARE MMTS MODELS CONSISTENTLY EFFECTIVE? (RQ1)

**MMTS models do not consistently outperform the strongest unimodal baselines.** Despite the growing enthusiasm around integrating text and time series for forecasting, our systematic evaluation in Table 1 reveals that simply incorporating textual context does *not* universally lead to better predictive performance. In fact, across our 16-dataset benchmark, unimodal models perform the best on 6 datasets in terms of MSE and 7 datasets in terms of MAE. We conduct a paired Student's *t*-test comparing unimodal and aligning-based Chronos, yielding *p*-values of 0.789 for MSE and 0.247 for MAE, indicating that the differences are not statistically significant. Due to space limits, we present additional results in Table 10 and Table 11 (Appendix D) for aligning-based methods, and in Table 17 and Table 20 (Appendix E) for prompting-based methods, which confirm the same trend. This suggests that MMTS methods do not universally lead to better performance, and the observed gains depend on specific modeling choices and data characteristics.

Table 1 also shows that aligning-based models tend to outperform prompting-based methods, achieving the lowest MSE and MAE on 12 and 9 datasets, compared to only 3 datasets on MSE and 2 datasets on MAE for prompting-based approaches. We attribute this at least partially to the limited numerical reasoning capabilities of current LLMs, which are primarily trained on text. As we will demonstrate in Section 4.2, increasing the capacity of the LLMs can improve performance, but a significant gap remains, compared to the strongest unimodal forecasting models.

Given that the added complexity of integrating text does not yield universal improvements in predictive accuracy, we conduct a deeper analysis from both architectural design and dataset characteristics, aiming to offer data-agnostic and generalizable guidelines on when multimodality helps.

---

[1]https://claude.ai/

Table 1: Main forecasting results across datasets. Models are grouped into: Unimodal (trained solely on time series), Aligning- (time-series + text alignment), and Prompting-based (directly prompting with pre-trained LLMs) methods. WQL and CRPS results can be found at Table 5 in Appendix. We also detail benchmarking results for more aligning- (Table 10, Table 11 in Appendix D) and prompting-based methods (Table 17, Table 20 in Appendix E). Winner/runner-up in green / blue .

| Dataset | Metric | Unimodal | | | Aligning | | | | Prompting | | | | | |
| | | PatchTST | DLinear | Chronos | PatchTST | DLinear | Chronos | Time-LLM | LLaMA-405B-Inst | Qwen-32B-Inst | Qwen-72B-Inst | Mistral 8×7B-Inst | GPT-4o (mini) | Claude-3.5 |
|---|---|---|---|---|---|---|---|---|---|---|---|---|---|---|
| Agriculture (Liu et al., 2024b) | MSE | 0.216 | 0.704 | 0.161 | 0.217 | 0.675 | 0.167 | 0.249 | 0.152 | 0.118 | 0.113 | 0.153 | 0.123 | 0.137 |
| | MAE | 0.304 | 0.598 | 0.258 | 0.315 | 0.577 | 0.299 | 0.336 | 0.269 | 0.229 | 0.227 | 0.253 | 0.243 | 0.244 |
| Climate (Liu et al., 2024b) | MSE | 1.142 | 0.956 | 1.200 | 1.147 | 0.949 | 1.077 | 1.127 | 1.913 | 1.468 | 1.265 | 1.786 | 1.519 | 1.733 |
| | MAE | 0.856 | 0.783 | 0.876 | 0.857 | 0.779 | 0.830 | 0.858 | 1.100 | 0.983 | 0.900 | 1.066 | 1.000 | 1.051 |
| Economy (Liu et al., 2024b) | MSE | 0.013 | 0.069 | 0.061 | 0.016 | 0.045 | 0.153 | 0.016 | 0.053 | 0.036 | 0.053 | 0.057 | 0.039 | 0.040 |
| | MAE | 0.091 | 0.210 | 0.200 | 0.099 | 0.166 | 0.344 | 0.103 | 0.179 | 0.153 | 0.184 | 0.186 | 0.162 | 0.159 |
| Energy (Liu et al., 2024b) | MSE | 0.108 | 0.098 | 0.112 | 0.107 | 0.097 | 0.138 | 0.102 | 0.131 | 0.125 | 0.139 | 0.144 | 0.137 | 0.131 |
| | MAE | 0.234 | 0.225 | 0.240 | 0.229 | 0.224 | 0.260 | 0.221 | 0.240 | 0.232 | 0.245 | 0.247 | 0.242 | 0.240 |
| Environment (Liu et al., 2024b) | MSE | 0.486 | 0.475 | 0.466 | 0.484 | 0.473 | 0.422 | 0.490 | 2.895 | 1.112 | 1.256 | 1.468 | 0.641 | 0.710 |
| | MAE | 0.500 | 0.531 | 0.486 | 0.499 | 0.530 | 0.488 | 0.499 | 0.913 | 0.699 | 0.788 | 0.807 | 0.568 | 0.592 |
| Health (Liu et al., 2024b) | MSE | 2.168 | 1.443 | 1.169 | 1.532 | 1.460 | 1.370 | 1.271 | 3.300 | 2.536 | 3.427 | 4.326 | 3.897 | 3.338 |
| | MAE | 0.842 | 0.809 | 0.676 | 0.761 | 0.821 | 0.833 | 0.734 | 1.096 | 0.947 | 1.045 | 1.100 | 1.218 | 1.218 |
| Socialgood (Liu et al., 2024b) | MSE | 0.863 | 1.030 | 0.914 | 0.863 | 1.013 | 0.957 | 1.019 | 1.078 | 0.810 | 0.678 | 0.866 | 0.908 | 0.788 |
| | MAE | 0.376 | 0.437 | 0.453 | 0.376 | 0.434 | 0.460 | 0.439 | 0.533 | 0.431 | 0.378 | 0.461 | 0.500 | 0.416 |
| Traffic (Liu et al., 2024b) | MSE | 0.147 | 0.153 | 0.182 | 0.147 | 0.152 | 0.180 | 0.162 | 0.419 | 0.262 | 0.275 | 0.359 | 0.311 | 0.232 |
| | MAE | 0.206 | 0.200 | 0.259 | 0.206 | 0.204 | 0.244 | 0.237 | 0.477 | 0.360 | 0.362 | 0.458 | 0.408 | 0.278 |
| Fashion (Skenderi et al., 2024) | MSE | 0.525 | 0.474 | 0.485 | 0.524 | 0.474 | 0.482 | 0.513 | 0.607 | 0.566 | 0.539 | 0.642 | 0.552 | 0.598 |
| | MAE | 0.523 | 0.515 | 0.468 | 0.523 | 0.516 | 0.472 | 0.511 | 0.508 | 0.487 | 0.486 | 0.550 | 0.480 | 0.507 |
| Weather (Kim et al., 2024) | MSE | 0.175 | 0.169 | 0.184 | 0.178 | 0.169 | 0.248 | 0.172 | 0.289 | 0.241 | 0.253 | 0.295 | 0.227 | 0.257 |
| | MAE | 0.315 | 0.314 | 0.325 | 0.319 | 0.314 | 0.379 | 0.315 | 0.397 | 0.360 | 0.373 | 0.403 | 0.358 | 0.374 |
| Medical (Kim et al., 2024) | MSE | 0.530 | 0.696 | 0.596 | 0.521 | 0.694 | 0.497 | 0.527 | 1.083 | 0.712 | 0.688 | 0.765 | 0.642 | 0.718 |
| | MAE | 0.514 | 0.624 | 0.565 | 0.511 | 0.624 | 0.503 | 0.513 | 0.603 | 0.572 | 0.571 | 0.601 | 0.554 | 0.575 |
| PTF (Wang et al., 2024) | MSE | 0.100 | 0.134 | 0.110 | 0.099 | 0.120 | 0.089 | 0.088 | 0.886 | 0.519 | 0.618 | 1.305 | 0.738 | 0.287 |
| | MAE | 0.173 | 0.227 | 0.167 | 0.174 | 0.215 | 0.162 | 0.189 | 0.670 | 0.479 | 0.538 | 0.808 | 0.647 | 0.323 |
| MSPG (Wang et al., 2024) | MSE | 0.324 | 0.415 | 0.365 | 0.363 | 0.408 | 0.300 | 0.353 | 2.635 | 1.165 | 2.731 | 2.879 | 1.488 | 0.873 |
| | MAE | 0.229 | 0.227 | 0.217 | 0.251 | 0.226 | 0.198 | 0.244 | 0.535 | 0.416 | 0.717 | 0.610 | 0.539 | 0.425 |
| LEU (Wang et al., 2024) | MSE | 0.504 | 0.497 | 0.522 | 0.504 | 0.495 | 0.513 | 0.490 | 1.543 | 0.822 | 0.811 | 1.185 | 0.890 | 0.787 |
| | MAE | 0.381 | 0.382 | 0.350 | 0.386 | 0.382 | 0.339 | 0.371 | 0.603 | 0.452 | 0.465 | 0.568 | 0.498 | 0.482 |
| MTFinance (Chen et al., 2025) | MSE | 0.003 | 0.002 | 0.002 | 0.060 | 0.002 | 0.004 | 0.005 | 0.006 | 0.002 | 0.004 | 0.015 | 0.002 | 0.002 |
| | MAE | 0.014 | 0.014 | 0.018 | 0.051 | 0.015 | 0.023 | 0.020 | 0.026 | 0.013 | 0.023 | 0.062 | 0.013 | 0.016 |
| MTWeather (Chen et al., 2025) | MSE | 0.210 | 0.203 | 0.209 | 0.204 | 0.202 | 0.220 | 0.187 | 2.179 | 0.623 | 0.846 | 3.481 | 0.558 | 2.110 |
| | MAE | 0.346 | 0.338 | 0.347 | 0.342 | 0.337 | 0.351 | 0.325 | 0.920 | 0.718 | 0.651 | 1.198 | 0.574 | 0.923 |
| Average | MSE | 0.470 | 0.470 | 0.421 | 0.435 | 0.464 | 0.426 | 0.423 | 1.198 | 0.695 | 0.856 | 1.233 | 0.792 | 0.796 |
| | MAE | 0.369 | 0.402 | 0.369 | 0.369 | 0.398 | 0.387 | 0.370 | 0.567 | 0.471 | 0.497 | 0.586 | 0.500 | 0.489 |

## 4.2 MODELING: DOES TEXT ENCODER CAPACITY AFFECT PERFORMANCE? (RQ2)

**Increasing text model capacity helps prompting-based methods, but a substantial gap remains compared to the best unimodal models.** We benchmark all the datasets in Table 1 as well as CiK (Williams et al., 2025), which is a small dataset that can be benchmarked through a zero-shot prompting instead of aligning approach (results found at Table 9 in Appendix). We observe that prompting-based methods benefit from scaling up the language models (Figure 3). Models with higher MMLU-Pro scores, indicative of stronger general reasoning abilities, tend to yield lower forecasting errors. Nevertheless, despite the significantly higher computational cost, even the strongest LLMs we evaluate do not match the unimodal Chronos model, shown as the red baseline in Figure 3.

**The improvement from larger LLMs stems from an enhanced numerical processing capabilities.** Given the previous results, we conduct experiments to ablate different text encoders from small to large ones in the aligning paradigm. As shown in Figure 4 (detailed numbers in Table 10 in Appendix D), we compare a range of text model architectures and sizes, including encoder-only models (BERT-base, BERT-large), encoder-decoder models (T5-base), and decoder-only models (GPT-2-small/medium/large). Somewhat non-intuitively, we find only slight correlations, with a large

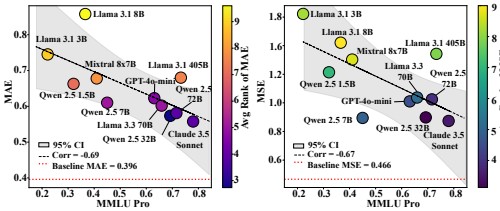 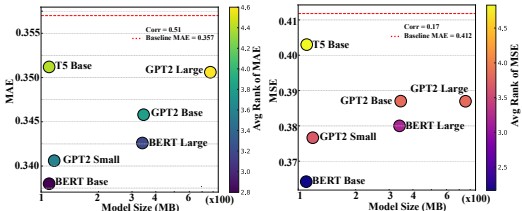

Figure 3: Prompting-based performance vs. MMLU-Pro. The y-axis shows MAE (left) and MSE (right), and the colormap shows the average ranking. Details in Table 1 and Table 20.

Figure 4: Aligning-based performance with varying text models. The y-axis shows MAE (left) and MSE (right), and the colormap shows the average ranking. Details in Table 10 in Appendix D.

dispersion between model size or architecture and forecasting performance. We also benchmark Qwen 1.5B with various fusion choices as well as different embeddings (Appendix D.2), which shows an inferior result than smaller models, such as BERT-base. We believe that such discrepancy between aligning- and prompting-based methods arises due to the differing roles of the text model in each paradigm. Prompting-based models must interpret both text and time series through the lens of the LLMs alone. Larger LLMs appear to have a greater capacity to process and parse numerical patterns within the time series data, a task smaller models struggle with without explicit guidance. As further shown in Section 4.3, textual information proves most helpful for smaller language models in the prompting-based method, indicating that smaller language models have difficulty understanding time series patterns without text. In contrast, large language models can extract relevant information from time series more effectively on their own. Aligning-based models, on the other hand, primarily rely on dedicated time series backbones that are specifically designed and excel at modeling temporal signals, rendering the text encoder's capacity less critical for capturing the time series dynamics itself.

**Reasoning models fail to improve the forecasting performance.** In our empirical evaluation, we employ distilled versions of the DeepSeek-R1 (DeepSeek-AI, 2025) architecture, specifically DeepSeek-R1-Distill-Qwen-1.5B and DeepSeek-R1-Distill-Qwen-7B. These models represent a fusion of the reasoning-centric training methodology of DeepSeek-R1 with the architectural foundations of the Qwen-1.5B and Qwen-7B. As shown in Figure 5, it is counterintuitive yet evident that the non-reasoning model outperforms the reasoning model distilled from DeepSeek-R1 across the majority of datasets for both Qwen 2.5 1.5B (blue bars) and Qwen 2.5 7B (red bars). As further illustrated in Appendix E.1, the reasoning

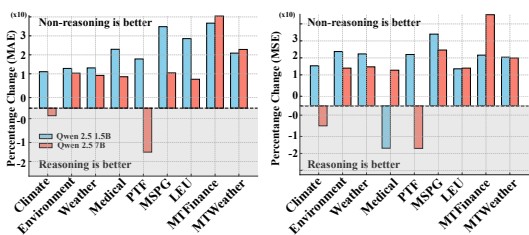

Figure 5: The percentage change in MAE and MSE when comparing non-reasoning models to reasoning models across 9 datasets. Positive values indicate non-reasoning models perform better. Details in Table 16 in Appendix E.

model tends to employ more simplistic methods and fails to effectively leverage the multimodality relationship within the data. This limitation is likely attributed to its reasoning capabilities being predominantly reinforced through textual information, with limited integration of temporal signals.

### 4.3 MODELING: DOES THE CAPACITY OF THE TIME SERIES MODEL MATTER? (RQ3)

**Extra modalities help especially weaker unimodal forecasting models.** For aligning-based MMTS methods, we observe a strong inverse relationship between unimodal forecasting strength and multimodal improvement. As shown in Figure 6, we evaluate a diverse set of time series backbones, including PatchTST, Informer, FEDformer, DLinear, iTransformer, Chronos, Time-LLM, TimeCMA and LeRet, and we find that models with higher unimodal forecasting errors tend to exhibit greater performance gains when enhanced with textual context. This suggests that text information provides the most value when the time series model lacks sufficient capacity to capture temporal patterns on its own. We detail the numbers of each time series model in Table 11 in Appendix D.

We observe that a similar trend holds for prompting-based approaches, as shown in Figure 7. Here, we define the unimodal forecasting model as prompting the LLM with only time series data, and

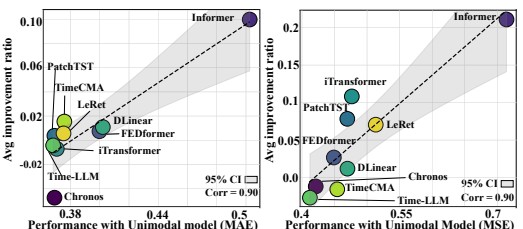 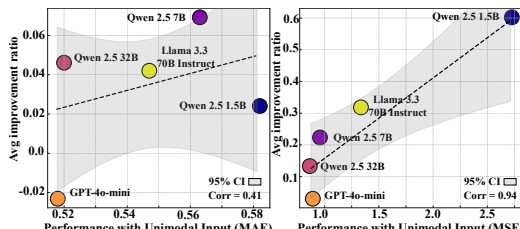

Figure 6: Aligning Performance gain with respect to unimodal time series model forecasting capability. Detailed results in Table 11 in Appendix D.

Figure 7: Prompting performance gain with respect to unimodal input forecasting capability. Detailed results in Table 21 in Appendix E.

we define the multimodal model as prompting the LLM with both time series and associated textual context. The experiment is performed on the few LLMs that yields a better result in Table 1. We find that lower-capacity LLMs, which struggle to interpret raw time series inputs effectively, benefit substantially more from the inclusion of textual guidance.

Together, these results suggest that the benefit of multimodal integration is conditional on the strength of the underlying unimodal forecasting model. When the model contains strong temporal inductive biases, additional textual information yields limited returns. However, for weaker unimodal models, text serves as a valuable auxiliary signal to compensate for shortcomings in time series understanding.

## 4.4 MODELING: HOW IMPORTANT IS THE DESIGN OF ALIGNING MECHANISMS? (RQ4)

Table 2: Different aligning choices: aggregation (add, concat), pooling (avg, CLS), projection (MLP, Residual), fusion location (early fusion, middle fusion, late fusion), and fine-tuning strategies (efficient, fixed, full, two-stage). Winner in each group in green ▪.

| Dataset | Metric | Uni | Aggregate | | Pooling | | Projector | | Location | | | Fine-tuning | | | |
| | | | Add | Concat | Avg | CLS | Residual | MLP | Early | Mid | Late | Efficient | Fixed | Full | Two-stage |
|---|---|---|---|---|---|---|---|---|---|---|---|---|---|---|---|
| Environment | MSE | 0.466 | **0.422** | 0.442 | 0.422 | **0.391** | 0.422 | **0.410** | **0.411** | 0.445 | 0.422 | 0.422 | 0.473 | 0.462 | **0.420** |
| | MAE | 0.486 | **0.488** | 0.504 | 0.488 | **0.464** | **0.488** | 0.489 | **0.467** | 0.496 | 0.488 | 0.488 | 0.488 | 0.500 | **0.483** |
| Medical | MSE | 0.596 | **0.497** | 0.585 | **0.497** | 0.570 | **0.497** | 0.523 | 0.703 | 0.559 | **0.497** | 0.497 | 0.538 | 0.617 | **0.483** |
| | MAE | 0.565 | **0.503** | 0.540 | **0.503** | 0.537 | **0.503** | 0.505 | 0.624 | 0.535 | **0.503** | 0.503 | 0.532 | 0.576 | **0.488** |
| PTF | MSE | 0.110 | 0.089 | **0.088** | 0.089 | **0.084** | 0.089 | **0.088** | 0.085 | **0.083** | 0.089 | 0.089 | 0.098 | 0.106 | **0.085** |
| | MAE | 0.167 | 0.162 | **0.158** | 0.162 | **0.150** | 0.162 | **0.160** | 0.163 | 0.166 | **0.162** | 0.162 | 0.157 | 0.162 | **0.150** |
| MSPG | MSE | 0.365 | **0.300** | 0.430 | **0.300** | 0.411 | **0.300** | 0.341 | 0.351 | 0.361 | **0.300** | **0.300** | 0.397 | 0.418 | 0.363 |
| | MAE | 0.217 | **0.198** | 0.230 | **0.198** | 0.229 | **0.198** | 0.204 | 0.205 | 0.209 | **0.198** | **0.198** | 0.229 | 0.223 | 0.217 |
| LEU | MSE | 0.522 | **0.513** | 0.551 | **0.513** | 0.546 | **0.513** | 0.519 | 0.515 | **0.500** | 0.513 | **0.513** | 0.526 | 0.531 | 0.568 |
| | MAE | 0.350 | **0.339** | 0.358 | **0.339** | 0.360 | **0.339** | 0.361 | 0.354 | 0.350 | **0.339** | **0.339** | 0.355 | 0.349 | 0.360 |
| **Average** | MSE | 0.412 | **0.364** | 0.419 | **0.364** | 0.400 | **0.364** | 0.376 | 0.413 | 0.390 | **0.364** | **0.364** | 0.406 | 0.427 | 0.384 |
| | MAE | 0.357 | **0.338** | 0.358 | **0.338** | 0.348 | **0.338** | 0.344 | 0.363 | 0.351 | **0.338** | **0.338** | 0.352 | 0.362 | 0.340 |

**Aligning paradigms affect performance, and optimal configurations are dataset-dependent.** For aligning-based methods, we conduct a comprehensive study of design choices for aligning textual and time series representations: (1) **Aggregate**: Combining text and time series embeddings via element-wise addition ("Add") or vector concatenation ("Concat"). Addition is generally more effective as it avoids feature explosion and overfitting; (2) **Pooling**: Extracting the text representation by mean pooling over token embeddings ("Avg") or selecting the [CLS] token ("CLS"), with average pooling better learning the global smoothed information from the context history; (3) **Projector**: Mapping text embeddings to the time series feature space using a multi-layer perceptron ("MLP") or a residual projection module ("residual"), with residual projection more robust as it preserves the original temporal information; (4) **Location**: Determining where to integrate text information within the forecasting architecture, with Chronos as example, before the time series encoder ("Early"), between encoder and decoder ("Mid"), or after the decoder ("Late"). Late fusion is the safest default because it minimizes disruption to temporal encoding at earlier stages; (5) **Fine-tuning**: Updating

Table 3: Controlled experiment on synthetic datasets.

| Model | Trend | | | | Seasonality | | | | Spike | | | |
|---|---|---|---|---|---|---|---|---|---|---|---|---|
| | Unique | | Redundant | | Unique | | Redundant | | Unique | | Redundant | |
| | MSE | MAE | MSE | MAE | MSE | MAE | MSE | MAE | MSE | MAE | MSE | MAE |
| Unimodal | 0.811 | 0.677 | 0.449 | 0.394 | 0.299 | 0.410 | 0.004 | 0.044 | 3.851 | 0.934 | 5.798 | 1.039 |
| Multimodal | 0.477 | 0.408 | 0.428 | 0.376 | 0.008 | 0.066 | 0.005 | 0.052 | 3.267 | 0.893 | 5.714 | 1.065 |

the text model fully ("full"), freezing ("fixed"), parameter-efficient tuning by only tuning the layer normalization layers ("efficient"), or using a two-stage procedure that first only tunes the projection head and then fine-tunes all the parameters ("two-stage"). Efficient fine-tuning is preferred for larger datasets with lower risk of underfitting, while two-stage fine-tuning is better for smaller datasets.

As summarized in Table 2, we find that different aligning choices influence forecasting performance. Although the best configuration varies across datasets, certain strategies consistently perform well on average: addition, average, residual projector, late fusion, and efficient fine-tuning. These consistent patterns suggest general guidelines for alignment design, highlighting principles of training stability, information preservation, and overfitting control.

### 4.5 DATA: HOW DO CONTEXT LENGTH AND TRAINING SAMPLE SIZE INFLUENCE PERFORMANCE? (RQ5)

**Aligning-based MMTS is more effective with larger training datasets.** As shown in Figure 8, we observe a positive correlation between dataset size and MMTS improvement across our real-world benchmark. This is novel and somewhat counterintuitive as one might expect multimodal models to gain relatively more in low-data regimes where textual context could compensate for limited training data. Instead, our results show that aligning-based MMTS yields greater relative improvements when more training data is available. This suggests that

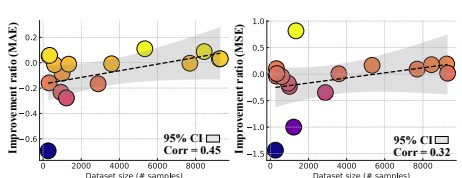

Figure 8: Improvement ratio vs dataset size. Each point represents a dataset from Table 4.

effective cross-modal representation learning requires sufficient data to fine-tune high-capacity multimodal models, particularly those with large text encoders. At the same time, for prompting-based methods in data-scarce zero-shot settings, textual context proves valuable, with multimodal prompting substantially outperforming unimodal prompting.

**Time series context length has limited effect on MMTS performance.** To study the role of temporal context, we down-sample the input lengths of five datasets with the longest context to 50% and 25% of its original size. For dynamic text, we apply the same down-sampling ratio; and for static text, we keep it unchanged. As shown in Figure 9, for aligning-based methods we find only slight correlation between time series context length and MMTS improvement with large dispersion. Similar observation holds for prompting-based methods as shown in Table 19. This could indicate that MMTS models primarily leverage text for contextual grounding, rather than for directly supplementing long-range temporal dependencies.

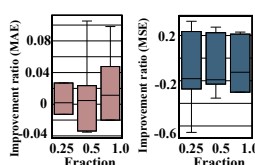

Figure 9: Reducing context length. Individual plots can be found at Table 14 in Appendix D.

### 4.6 DATA: HOW DOES TEXT AND TIME SERIES ALIGNMENT QUALITY AFFECT PERFORMANCE (RQ6)

**MMTS provides the greatest benefit when text contains predictive information that is complementary to the time series.** We construct a synthetic multimodal benchmark in which we explicitly control the relationship between text and time series. This controlled study is designed to be an oracle experiment, not a realistic forecasting scenario, and is intended to analyze model behavior under various controlled conditions rather than to introduce information leakage. Specifically, we simulate a time series using Gaussian processes with changepoints in the *trend*, and we define two conditions: (1) **unique-text**, where the changepoint occurs at the forecasting boundary and the direction of change is provided only in the text, and (2) **redundant-text**, where the

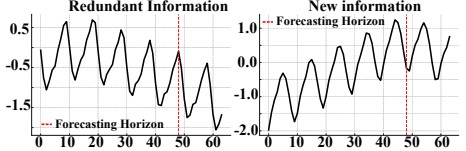

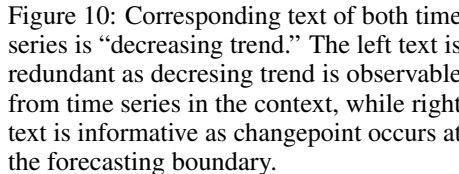

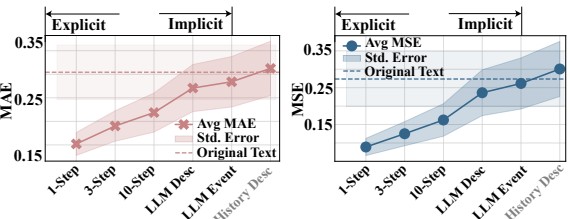

Figure 10: Corresponding text of both time series is "decreasing trend." The left text is redundant as decresing trend is observable from time series in the context, while right text is informative as changepoint occurs at the forecasting boundary.

Figure 11: MMTS shows larger improvements when the text information is more explicit. This plot shows an average results of 9 datasets, and individual plots of each datasets can be found at Figure 18 in Appendix D.

changepoint occurs within the observed context, and thus can be inferred from the time series alone. We provide an example in Figure 10, where both plots have associated text "decreasing trend." The text is redundant in the left figure as the decreasing trend is already observable from the context time series, while it is informative to the right figure, as the decreasing trend happens at the changepoint. As shown in Table 3, MMTS significantly outperforms unimodal models in the unique-text setting, but it shows no meaningful advantage when the text information is redundant. We observe similar patterns when altering time series properties such as *frequency* and *spikes*, as shown in Table 3. We detail the synthetic dataset generation in Appendix F. This shows a general and data-agnostic conclusion that MMTS gains are substantial only when text introduces novel, predictive signals not observable in the time series context. This finding aligns with a previous study from CiK(Williams et al., 2025) based in zero-shot setting. Our analysis also complements CiK: whereas CiK provides fixed, real-world paired text for evaluating whether text helps, our synthetic benchmark enables controlled variants to isolate why and when text helps, offering scalability for aligning-based models.

**Explicit text contributes more.** We validate our above finding in real-world datasets. As shown in Figure 11, we create several variants of text from the real-world datasets based on future information: (1) **"1-Step"**: the exact change ratio at the next time point; (2) **"3-Step"** and **"10-Step"**: the average change over the next 3 or 10 steps; (3) **"LLM Desc"**: natural language descriptions generated by prompting Claude 3.5 Sonnet on the future time series; (4) **"LLM Event"**: LLM-generated explanations of potential events underlying future trends and variations. We provide more details and examples for these text variants in Appendix G. These variants progressively reduce the alignment quality between text and target signal by varying the explicitness of the text. We find that forecasting performance degrades as the informativeness of the text decreases, confirming that MMTS is effective only when the text modality adds unique predictive value. We also evaluate a controlled setting where the LLM is prompted to generate descriptions of the past time series (**"History Desc"**). This historical text contains no new information beyond what is already encoded in the time series input, and yields the highest forecasting error among all variants.

We compare these synthetic text data with real text within the benchmark and overlay MMTS errors using the original textual descriptions as horizontal lines. We present the aggregated results on 9 datasets in Figure 11 and show their individual plots in Figure 18. For datasets where the real-text error lines are low, such as MSPG, LEU, PTF, Medical, and Environment, we also observe consistent MMTS gains in Table 1, suggesting that these datasets feature high-quality text-and-time-series alignment. These results highlight MMTS models are most effective when the text provides information that is both relevant and complementary to the time series signal.

## 5 CONCLUSION AND DISCUSSION

In this work, we provide new insights into when and how multimodality improves forecasting performance. Contrary to the (sometimes implicit) assumptions prevailing in recent work, our study finds that MMTS methods do not consistently outperform the strongest unimodal baselines. We provide a rigorous and quantitative analysis on their effectiveness with respective to model capacity, alignment strategy, and data characteristics. Through controlled and data-agnostic experiments, we provide more general guidelines to future MMTS research that extends beyond the current benchmark. Our work helps clarify when MMTS forecasting is truly beneficial, encouraging more transparent and data-aware modeling practices. Reproducibility details are discussed in Appendix B and Appendix C.

## REPRODUCIBILITY STATEMENT

Appendix A provides detailed descriptions of datasets used in this work. Appendix B outlines the implementation details including hyperparameters for aligning-based methods, and Appendix C details the prompting-based methods including the specific prompts. Appendix F describes the procedure for synthetic time-series generation, and Appendix G details synthetic text generation.

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

CONTENTS

LIST OF FIGURES

LIST OF TABLES

# A    DATASETS

We provide the statistics of each dataset in Table 4.

For datasets with predefined training, validation, and test splits, we adopt their original partitions. For datasets without publicly available splits, we use the last 20% of the data as the test set, use the last one-seventh of the remaining data for validation, and the rest for training. For the Fashion dataset, we use data from the Autumn-Winter 2017, Autumn-Winter 2018, Spring-Summer 2017, and Spring-Summer 2018 seasons for training and validation (with the last one-seventh used for validation), and data from the Autumn-Winter 2019 and Spring-Summer 2019 seasons for testing.

Each dataset is standardized using the mean and standard deviation computed from its training set, and we report the errors in the normalized space. We observed that some baseline implementations discard samples from the final batch; for consistency, we keep all data including those in the last batch across all evaluated methods. Additionally, certain datasets include prior statistics about the underlying data distributions. To ensure a fair and uniform evaluation across datasets, and to isolate the contribution of text information, we exclude such prior statistics during training and evaluation.

Table 4: Dataset statistics.

| Dataset Name | Type | Train Size | Val Size | Test Size | Context Length | Prediction Length |
|---|---|---|---|---|---|---|
| Agriculture (Liu et al., 2024b) | Dynamic | 318 | 45 | 94 | 24 | 6 |
| Climate (Liu et al., 2024b) | Dynamic | 318 | 45 | 94 | 24 | 6 |
| Economy (Liu et al., 2024b) | Dynamic | 267 | 38 | 79 | 24 | 6 |
| Energy (Liu et al., 2024b) | Dynamic | 1000 | 138 | 284 | 24 | 12 |
| Environment (Liu et al., 2024b) | Dynamic | 7700 | 1064 | 2173 | 24 | 48 |
| Health (Liu et al., 2024b) | Dynamic | 937 | 129 | 266 | 24 | 12 |
| Socialgood (Liu et al., 2024b) | Dynamic | 601 | 85 | 175 | 24 | 6 |
| Traffic (Liu et al., 2024b) | Dynamic | 342 | 49 | 101 | 24 | 6 |
| Fashion (Skenderi et al., 2024) | Static | 3081 | 513 | 1983 | 1 | 11 |
| Weather (Kim et al., 2024) | Dynamic | 2475 | 412 | 361 | 7 | 7 |
| Medical (Kim et al., 2024) | Dynamic | 4575 | 762 | 706 | 7 | 7 |
| PTF (Wang et al., 2024) | Static | 7927 | 1321 | 2272 | 120 | 24 |
| MSPG (Wang et al., 2024) | Static | 7244 | 1207 | 2106 | 480 | 96 |
| LEU (Wang et al., 2024) | Static | 7968 | 1328 | 2320 | 240 | 48 |
| MTFinance (Chen et al., 2025) | Static | 1224 | 204 | 356 | 276 | 78 |
| MTWeather (Chen et al., 2025) | Static | 1344 | 224 | 391 | 336 | 72 |

Table 5: CRPS and WQL Results for Table 1.

| Dataset | Metric | Unimodal | Aligning | LLaMA-405B | Qwen-32B | Qwen-72B | Mistral 8×7B | GPT-4o(mini) | Claude 3.5 |
|---|---|---|---|---|---|---|---|---|---|
| Agriculture (Liu et al., 2024b) | CRPS | 0.190 | 0.198 | 0.236 | 0.212 | 0.198 | 0.253 | 0.209 | 0.219 |
| | WQL | 0.082 | 0.084 | 0.083 | 0.071 | 0.070 | 0.081 | 0.074 | 0.071 |
| Climate (Liu et al., 2024b) | CRPS | 0.632 | 0.528 | 0.851 | 0.815 | 0.708 | 1.066 | 0.800 | 0.910 |
| | WQL | 0.889 | 0.791 | 1.304 | 1.189 | 1.045 | 1.368 | 1.191 | 1.289 |
| Economy (Liu et al., 2024b) | CRPS | 0.136 | 0.244 | 0.149 | 0.139 | 0.150 | 0.186 | 0.133 | 0.136 |
| | WQL | 0.061 | 0.106 | 0.058 | 0.051 | 0.059 | 0.064 | 0.052 | 0.053 |
| Energy (Liu et al., 2024b) | CRPS | 0.166 | 0.179 | 0.206 | 0.212 | 0.215 | 0.247 | 0.209 | 0.217 |
| | WQL | 0.210 | 0.229 | 0.574 | 0.541 | 0.527 | 0.602 | 0.548 | 0.561 |
| Environment (Liu et al., 2024b) | CRPS | 0.321 | 0.305 | 0.627 | 0.446 | 0.486 | 0.807 | 0.402 | 0.475 |
| | WQL | 0.540 | 0.519 | 1.336 | 0.826 | 0.947 | 1.078 | 0.707 | 0.766 |
| Health (Liu et al., 2024b) | CRPS | 0.578 | 0.667 | 0.861 | 0.836 | 0.857 | 1.100 | 1.031 | 1.067 |
| | WQL | 0.647 | 0.777 | 1.335 | 1.252 | 1.347 | 1.543 | 1.608 | 1.722 |
| Socialgood (Liu et al., 2024b) | CRPS | 0.353 | 0.337 | 0.454 | 0.385 | 0.313 | 0.461 | 0.411 | 0.370 |
| | WQL | 0.346 | 0.344 | 0.600 | 0.558 | 0.410 | 0.580 | 0.609 | 0.471 |
| Traffic (Liu et al., 2024b) | CRPS | 0.199 | 0.202 | 0.390 | 0.298 | 0.281 | 0.458 | 0.344 | 0.239 |
| | WQL | 0.149 | 0.147 | 0.335 | 0.251 | 0.252 | 0.329 | 0.287 | 0.204 |
| Fashion (Skenderi et al., 2024) | CRPS | 0.290 | 0.296 | 0.493 | 0.466 | 0.448 | 0.550 | 0.441 | 0.458 |
| | WQL | 0.665 | 0.680 | 1.075 | 0.998 | 0.964 | 1.157 | 0.954 | 0.981 |
| Weather (Kim et al., 2024) | CRPS | 0.223 | 0.268 | 0.317 | 0.305 | 0.304 | 0.403 | 0.295 | 0.315 |
| | WQL | 0.350 | 0.412 | 0.677 | 0.624 | 0.636 | 0.729 | 0.611 | 0.631 |
| Medical (Kim et al., 2024) | CRPS | 0.382 | 0.339 | 0.496 | 0.500 | 0.472 | 0.601 | 0.465 | 0.505 |
| | WQL | 0.371 | 0.328 | 0.581 | 0.579 | 0.554 | 0.644 | 0.547 | 0.580 |
| PTF (Wang et al., 2024) | CRPS | 0.140 | 0.135 | 0.423 | 0.331 | 0.361 | 0.476 | 0.477 | 0.280 |
| | WQL | 0.197 | 0.191 | 0.899 | 0.568 | 0.642 | 0.770 | 0.770 | 0.436 |
| MSPG (Wang et al., 2024) | CRPS | 0.182 | 0.173 | 0.361 | 0.383 | 0.309 | 0.611 | 0.371 | 0.338 |
| | WQL | 0.347 | 0.323 | 0.718 | 0.650 | 0.695 | 0.884 | 0.769 | 0.650 |
| LEU (Wang et al., 2024) | CRPS | 0.240 | 0.239 | 0.347 | 0.329 | 0.309 | 0.568 | 0.362 | 0.337 |
| | WQL | 0.479 | 0.474 | 0.859 | 0.674 | 0.695 | 0.941 | 0.719 | 0.792 |
| MTFinance (Chen et al., 2025) | CRPS | 0.012 | 0.016 | 0.017 | 0.012 | 0.017 | 0.262 | 0.012 | 0.013 |
| | WQL | 0.023 | 0.031 | 0.172 | 0.061 | 0.459 | 1.170 | 0.067 | 0.091 |
| MTWeather (Chen et al., 2025) | CRPS | 0.236 | 0.234 | 0.641 | 0.504 | 0.423 | 1.198 | 0.451 | 0.527 |
| | WQL | 0.368 | 0.370 | 1.598 | 1.054 | 1.074 | 2.223 | 0.877 | 1.371 |
| **Average** | CRPS | 0.267 | 0.273 | 0.429 | 0.386 | 0.366 | 0.578 | 0.401 | 0.400 |
| | WQL | 0.358 | 0.363 | 0.763 | 0.622 | 0.649 | 0.885 | 0.649 | 0.667 |

## B    IMPLEMENTATION DETAILS OF ALIGNING-BASED METHODS

**Experiment compute resources.** To train and evaluate aligning-based models, we use eight 40GB A100 GPUs. Our implementation is based on PYTORCH.

**Hyperparameters.** We implemented the Chronos-based MMTS model by aligning embeddings from various text encoders to those of Chronos-Bolt[2], a patching-based variant of the original Chronos model built on a T5 encoder-decoder architecture. Alignment strategies are compared in Section 4.4 and the default aligning strategy is the best configuration found in Section 4.4 (addition, average, residual projector, late fusion, and efficient fine-tuning). Text encoder choices are evaluated in Section 4.2. The default text encoder is BERT-Base, and we use a learning rate of 0.001 and train with the AdamW optimizer, a batch size of 32 per GPU, and 4 GPUs in total. Early stopping is applied based on validation performance with a patience of 50 steps. The projection module consists of a residual block with a hidden layer mapping the text embedding dimension (e.g., 768 for BERT-Base) to 2048, followed by an output layer projecting to 512, and a residual connection mapping directly from the text embedding dimension to 512.

For aligning-based models utilizing Informer, FEDformer, PatchTST, iTransformer, and DLinear as the time series backbones, we adopt the official implementations provided by MMTSF-Lib (Liu et al., 2024b)[3]. To fuse textual and temporal information, we vary the ratio at which text embeddings are added to the time series representations from 0.1 to 0.5, selecting the optimal value for each model individually. We report the hyper-parameter tuning results in Table 6. We tune from 0.1 as values closer to 0.0 effectively reduce the model to a unimodal setting (see the 0.0 column, which performs comparably to the unimodal baseline). We tune up to 0.5 as we already observed a clear trend of performance degradation when we increase the values up to 0.5. The projector is trained with a learning rate of 0.01, while the time series model is optimized with a learning rate of 0.0001. We use a pretrained BERT model with frozen parameters, extracting its input embeddings to serve as the text representations. We use the official implementations for Time-LLM[4], TimeCMA[5] and LeRet[6], with a learning rate of 0.01, 0.001, and 0.005 respectively. We replace their language embeddings with zero embeddings for the corresponding unimodal counterparts.

Table 6: Hyper-parameter tuning on the text embedding ratio (using DLinear as an example).

| Dataset | Unimodal | | 0.0 | | 0.1 | | 0.2 | | 0.3 | | 0.4 | | 0.5 | |
|---|---|---|---|---|---|---|---|---|---|---|---|---|---|---|
| Metric | MSE | MAE | MSE | MAE | MSE | MAE | MSE | MAE | MSE | MAE | MSE | MAE | MSE | MAE |
| Agriculture | 0.704 | 0.598 | 0.685 | 0.59 | 0.675 | 0.577 | 0.669 | 0.581 | 0.687 | 0.608 | 0.73 | 0.647 | 0.802 | 0.694 |
| Climate | 0.956 | 0.783 | 0.969 | 0.789 | 0.949 | 0.779 | 0.949 | 0.778 | 0.964 | 0.786 | 0.98 | 0.793 | 0.999 | 0.799 |
| Economy | 0.069 | 0.21 | 0.07 | 0.21 | 0.045 | 0.166 | 0.048 | 0.173 | 0.077 | 0.229 | 0.134 | 0.311 | 0.219 | 0.41 |
| Energy | 0.098 | 0.225 | 0.096 | 0.221 | 0.097 | 0.224 | 0.107 | 0.236 | 0.11 | 0.24 | 0.099 | 0.226 | 0.102 | 0.23 |
| Environment | 0.475 | 0.531 | 0.474 | 0.531 | 0.473 | 0.53 | 0.473 | 0.53 | 0.473 | 0.53 | 0.474 | 0.53 | 0.473 | 0.53 |
| Health | 1.443 | 0.809 | 1.443 | 0.809 | 1.46 | 0.821 | 1.47 | 0.821 | 1.487 | 0.83 | 1.494 | 0.833 | 1.475 | 0.827 |
| Socialgood | 1.03 | 0.437 | 1.045 | 0.441 | 1.013 | 0.434 | 0.992 | 0.43 | 0.968 | 0.425 | 0.938 | 0.42 | 0.902 | 0.413 |
| Traffic | 0.153 | 0.2 | 0.152 | 0.21 | 0.152 | 0.204 | 0.152 | 0.203 | 0.151 | 0.206 | 0.152 | 0.212 | 0.155 | 0.208 |
| Fashion | 0.474 | 0.515 | 0.476 | 0.517 | 0.474 | 0.516 | 0.478 | 0.518 | 0.487 | 0.523 | 0.489 | 0.525 | 0.492 | 0.527 |
| Weather | 0.169 | 0.314 | 0.169 | 0.314 | 0.169 | 0.314 | 0.169 | 0.314 | 0.169 | 0.314 | 0.169 | 0.314 | 0.169 | 0.314 |
| Medical | 0.696 | 0.624 | 0.694 | 0.623 | 0.694 | 0.624 | 0.693 | 0.623 | 0.694 | 0.623 | 0.693 | 0.623 | 0.694 | 0.623 |
| PTF | 0.134 | 0.227 | 0.134 | 0.226 | 0.12 | 0.215 | 0.129 | 0.226 | 0.118 | 0.217 | 0.117 | 0.218 | 0.123 | 0.218 |
| MSPG | 0.415 | 0.227 | 0.412 | 0.226 | 0.408 | 0.226 | 0.415 | 0.227 | 0.421 | 0.229 | 0.408 | 0.23 | 0.396 | 0.243 |
| LEU | 0.497 | 0.382 | 0.495 | 0.38 | 0.495 | 0.382 | 0.495 | 0.381 | 0.494 | 0.381 | 0.496 | 0.382 | 0.496 | 0.384 |
| MTFinance | 0.002 | 0.014 | 0.003 | 0.018 | 0.002 | 0.015 | 0.002 | 0.016 | 0.004 | 0.022 | 0.002 | 0.016 | 0.003 | 0.018 |
| MTWeather | 0.203 | 0.338 | 0.2 | 0.336 | 0.202 | 0.337 | 0.199 | 0.336 | 0.2 | 0.338 | 0.201 | 0.34 | 0.202 | 0.341 |
| **Avg** | 0.470 | 0.402 | 0.470 | 0.403 | 0.464 | 0.398 | 0.465 | 0.400 | 0.469 | 0.406 | 0.474 | 0.414 | 0.481 | 0.424 |

---

[2] https://huggingface.co/autogluon/chronos-bolt-small
[3] https://github.com/AdityaLab/MM-TSFlib
[4] https://github.com/KimMeen/Time-LLM
[5] https://github.com/ChenxiLiu-HNU/TimeCMA
[6] https://github.com/hqh0728/LeRet

## C  IMPLEMENTATION DETAILS OF PROMPTING-BASED METHODS

### C.1  LARGE LANGUAGE MODELS (LLMs) BENCHMARKED

We benchmarked a total of 11 LLMs as illustrated in Figure 3. These models include various versions of LLaMA, Qwen, and GPT-4o-mini, as well as Claude 3.5 Sonnet and Mixtral 8x7B Instruct. Additionally, we included DeepSeek 1.5B and DeepSeek 7B in our evaluations. For prompting-based methods, the major hyper-parameter is the temperature, which is optimized following CiK.

The models are sourced from the following platforms:

- **LLaMA (3B, 8B, 405B)**: All LLaMA models were called directly from **Amazon Bedrock**.
- **Qwen (2.5 1.5B, 2.5 7B, 2.5 32B, 2.5 72B)**: The Qwen models were downloaded from **Huggingface**.
    - Qwen2.5-1.5B-Instruct
    - Qwen2.5-7B-Instruct
    - Qwen2.5-32B-Instruct
    - Qwen2.5-72B-Instruct
- **GPT-4o-mini**: Sourced from OpenAI API.
- **Claude 3.5 Sonnet**: Also deployed via Amazon Bedrock.
- **Mixtral-8x7B-Instruct-v0.1**: Accessed through **Huggingface**.
- **DeepSeek (1.5B, 7B)**: Downloaded from **Huggingface**.
    - DeepSeek-R1-Distill-Qwen-1.5B
    - DeepSeek-R1-Distill-Qwen-7B

### C.2  PROMPTING METHODOLOGY

We employed two distinct prompting strategies depending on the availability of contextual text information. When contextual text was available, we used the following prompt:

---

**Prompt for time series forecasting with multimodal input**

```
# Context-Informed Time Series Forecasting
## Background Information
<context>
{context}
</context>
## Historical Data
<history>
{history}
</history>
## Task Description
You are a specialized forecasting agent tasked with predicting
the next {PREDICTION_LENGTH} timestamps in this time series.
Your forecast should:
1. Leverage patterns identified in the historical data
2. Incorporate relevant contextual information
3. Account for any seasonality, trends, or cyclical patterns
4. Consider how external factors described in the context might
influence future values
## Analytical Approach
- First, analyze the historical data to identify
baseline patterns, trends, and seasonality
- Second, examine the provided context to understand factors
that may influence the forecast
- Third, determine how these contextual factors might modify
```

---

```
expected patterns
- Finally, generate precise predictions that integrate both
historical patterns and contextual insights
## Output Format Requirements
Return your forecast using exactly this format:
<forecast>
index_1, value_1
index_2, value_2
...
index_{PREDICTION_LENGTH}, value_{PREDICTION_LENGTH}
</forecast>
Important:
- Each forecast entry must appear on its own line
- Use ONLY the (index, value) format within the forecast tags
- Maintain consistent numerical precision with the historical
data
- DO NOT include any explanations, notes, or reasoning between
the forecast tags
- Ensure predictions properly reflect both the historical
patterns and contextual factors
```

When contextual text was not available, we employed the following streamlined prompt:

**Prompt for time series forecasting with unimodal input**

```
Time Series Forecasting Task
## Historical Data
<history>
{history}
</history>
## Task Description
You are a specialized time series forecasting agent. Your goal
is to predict the next {PREDICTION_LENGTH} timestamps based on
the historical data provided above.
## Data Analysis Instructions
1. Analyze the pattern, trend, and seasonality in the
historical data
2. Identify any recurring cycles or anomalies
3. Consider both recent and long-term patterns in your
prediction
4. Account for any contextual factors evident in the data
## Output Format Requirements
Return your forecast in a structured (index, value) format
within <forecast></forecast> tags as follows:
<forecast>
index_1, value_1
index_2, value_2
...
index_n, value_n
</forecast>
Important:
- Each prediction should be on a new line
- Include ONLY the index and predicted value pairs within the
tags
- Do NOT include explanations, reasoning, or additional text
between the tags
- Ensure numeric precision appropriate to the historical data
```

```
  - Maintain consistent units and scale with the historical data
```

Performance metrics were parsed directly from the `<forecast></forecast>` tags within the LLM's output. For each time series trace, we employed the setting `Sample = True` to generate three distinct forecast runs. The uncertainty quantification metrics presented in the tables are derived from these three runs.

**Experiment compute resources.** The experiments involving the Huggingface-hosted large language models were conducted utilizing a compute infrastructure equipped with 8 NVIDIA A100 GPUs, each with 40GB of memory.

## C.3 PROMPT OPTIMIZATION

We perform prompt optimization based on the following prompt from CiK (with a slight modification since most of the datasets do not have constraints):

---

**Original prompt for time series forecasting with multimodal input**

```
I have a time series forecasting task for you.

<context>
{context}
</context>

<history>
{history}
</history>

Please predict the next {PREDICTION_LENGTH} timestamps based
on the context.

Return the forecast in (index, value) format, enclosed within
the <forecast> and </forecast> tags. Separate each forecast
with a new line.
Each line should contain the index and the corresponding
forecasted
value. Do not include any explanations between the <forecast>
and </
forecast> tags.
```

---

To enhance the performance of prompting-based methods, we conducted a series of prompt optimization which leads to our prompt in Section C.2. We evaluated the impact of these optimizations on several key metrics: MAE, MSE, CRPS and WQL. The results obtained using the original prompt (Table 7) were compared against those achieved with the optimized prompt (Table 8). Following the optimization process, the results demonstrated notable improvements across several domains and metrics. More specifically, the improved prompt significantly reduced errors for environmental data (>50% MSE decrease) and showed modest gains for health, agriculture and economy datasets, though it produced mixed or slightly negative outcomes for other domains.

## C.4 REPRODUCING CONTEXT IS KEY (CIK)

We reproduce the results of CiK(Williams et al., 2025) to examine how LLM performance scales on MMLU-PRO. In addition to the Continuous Ranked Probability Score (CRPS), we report Mean Absolute Error (MAE) and Mean Squared Error (MSE) for consistency with the rest of this paper, using the authors' public code[7]. Because CiK caps the Risk-Conditioned CRPS (RCRPS) at 5 to keep extreme failures from distorting the aggregate score, we apply the same 5-point cap to MAE and MSE.

---

[7]`https://github.com/ServiceNow/context-is-key-forecasting`

Table 7: Performance metrics for the original prompt from CiK.

| Domain | MAE | MSE | WQL | CRPS |
|---|---|---|---|---|
| Agriculture | 0.23889 | 0.12689 | 0.07351 | 0.22454 |
| Climate | 0.94299 | 1.33173 | 1.09906 | 0.80673 |
| Economy | 0.14656 | 0.03381 | 0.04876 | 0.13578 |
| Environment | 0.80476 | 1.71238 | 0.89888 | 0.52064 |
| Energy | 0.22635 | 0.12097 | 0.52654 | 0.21158 |
| Health | 0.96483 | 4.03459 | 1.15122 | 0.86772 |
| Socialgood | 0.41672 | 0.79210 | 0.53083 | 0.38797 |
| Traffic | 0.36918 | 0.27783 | 0.25850 | 0.33231 |
| Average | 0.51379 | 1.05379 | 0.57341 | 0.43591 |

Table 8: Performance metrics for the optimized prompt.

| Domain | MAE | MSE | WQL | CRPS |
|---|---|---|---|---|
| Agriculture | 0.22782 | 0.11962 | 0.07016 | 0.21137 |
| Climate | 0.96669 | 1.40956 | 1.13980 | 0.80608 |
| Economy | 0.14503 | 0.03252 | 0.04775 | 0.13017 |
| Environment | 0.64069 | 0.90954 | 0.75110 | 0.45014 |
| Energy | 0.23310 | 0.12990 | 0.53894 | 0.21454 |
| Health | 0.99145 | 2.95664 | 1.28843 | 0.88181 |
| Socialgood | 0.44685 | 0.82796 | 0.57712 | 0.40388 |
| Traffic | 0.38470 | 0.29720 | 0.26979 | 0.32981 |
| Average | 0.50454 | 0.83537 | 0.58539 | 0.42848 |

Most datasets (Table 4) in our study lack a designated region of interest, so we omit RCRPS and instead present standard CRPS, Weighted Quantile Loss (WQL), MSE, and MAE. Our CRPS values closely match those reported in the original CiK paper, again highlighting `Llama-405B` as the top-performing model on their benchmarks (Table 9).

Table 9: Model performance metrics of CiK (MAE, MSE, WQL, CRPS).

| Model | MAE | MSE | WQL | CRPS |
|---|---|---|---|---|
| Qwen 32B | 1.803 | 2.981 | 1.316 | 1.517 |
| Qwen 72B | 1.439 | 2.522 | 0.978 | 1.133 |
| GPT 4o-mini | 1.924 | 3.042 | 1.363 | 1.630 |
| Llama 405B | 2.079 | 3.157 | 0.774 | 0.192 |
| Mixtral-8x7B-Instruct-v0.1 | 2.052 | 3.311 | 1.761 | 0.528 |
| Qwen7B | 1.812 | 2.997 | 1.415 | 1.561 |
| Qwen 1.5B | 1.982 | 3.024 | 1.499 | 1.556 |
| llama 3b | 1.640 | 2.839 | 1.079 | 0.299 |
| llama 8b | 2.077 | 3.189 | 1.079 | 0.299 |
| llama 70 | 1.695 | 2.825 | 0.822 | 0.314 |
| Claude 3.5 | 1.480 | 2.474 | 0.950 | 1.246 |

Finally, we observed that Amazon Bedrock returns only one completion per invocation—even with `sample=True`. To obtain multiple samples, we therefore issue repeated calls to the LLM endpoint.

# D ADDITIONAL RESULTS FOR ALIGNING-BASED METHODS

## D.1 PERFORMANCE WITH DIFFERENT TEXT MODELS AND TIME SERIES MODELS

Table 10: Aligning performance with different text models: BERT-Base, BERT-Large, T5-Base, GPT2-Small, GPT2-Base, GPT2-Large. Aggregated results shown in Figure 4.

| Dataset | Metric | Unimodal | BERT-Base | BERT-Large | T5-Base | GPT2-Small | GPT2-Base | GPT2-Large |
|---|---|---|---|---|---|---|---|---|
| Environment | MSE | 0.466 | 0.422 | 0.414 | 0.463 | 0.467 | 0.462 | 0.443 |
| | MAE | 0.486 | 0.488 | 0.488 | 0.500 | 0.485 | 0.489 | 0.514 |
| Medical | MSE | 0.596 | 0.497 | 0.521 | 0.542 | 0.497 | 0.479 | 0.535 |
| | MAE | 0.565 | 0.503 | 0.505 | 0.527 | 0.502 | 0.497 | 0.518 |
| PTF | MSE | 0.110 | 0.089 | 0.088 | 0.083 | 0.084 | 0.081 | 0.081 |
| | MAE | 0.167 | 0.162 | 0.157 | 0.154 | 0.162 | 0.155 | 0.156 |
| MSPG | MSE | 0.365 | 0.300 | 0.362 | 0.396 | 0.305 | 0.372 | 0.339 |
| | MAE | 0.217 | 0.198 | 0.212 | 0.216 | 0.193 | 0.228 | 0.212 |
| LEU | MSE | 0.522 | 0.513 | 0.515 | 0.529 | 0.531 | 0.541 | 0.536 |
| | MAE | 0.350 | 0.339 | 0.351 | 0.359 | 0.361 | 0.360 | 0.353 |

We detail the performance of aligning-based MMTS given different text models (BERT-Base, BERT-Large, T5-Base, GPT2-Small, GPT2-Base, GPT2-Large) in Table 10. Aggregated results across datasets are presented in Figure 4 of the main text.

Table 11: Unimodal and aligning-based MMTS performance with varying time series models: Chronos, Informer, FEDformer, PatchTST, iTransformer, DLinear, TimeLLM, TimeCMA, LeRet. Aggregated results in Figure 6.

| Dataset | Metric | Unimodal | | | | | | | | | MMTS | | | | | | | | |
|---|---|---|---|---|---|---|---|---|---|---|---|---|---|---|---|---|---|---|---|
| | | Chronos | Informer | FEDformer | PatchTST | iTransformer | DLinear | Time-LLM | TimeCMA | LeRet | Chronos | Informer | FEDformer | PatchTST | iTransformer | DLinear | Time-LLM | TimeCMA | LeRet |
| Agriculture | MSE | 0.161 | 3.797 | 0.214 | 0.216 | 0.176 | 0.704 | 0.223 | 0.215 | 0.186 | 0.167 | 1.830 | 0.160 | 0.217 | 0.181 | 0.675 | 0.169 | 0.169 | 0.215 |
| | MAE | 0.258 | 1.537 | 0.311 | 0.304 | 0.281 | 0.598 | 0.317 | 0.300 | 0.304 | 0.299 | 1.041 | 0.283 | 0.315 | 0.287 | 0.577 | 0.336 | 0.278 | 0.314 |
| Climate | MSE | 1.200 | 1.250 | 1.130 | 1.142 | 1.115 | 0.956 | 1.000 | 1.375 | 1.911 | 1.077 | 1.290 | 1.155 | 1.147 | 1.109 | 0.949 | 1.127 | 1.247 | 1.516 |
| | MAE | 0.876 | 0.910 | 0.872 | 0.856 | 0.845 | 0.783 | 0.811 | 0.950 | 1.093 | 0.830 | 0.919 | 0.882 | 0.857 | 0.842 | 0.779 | 0.858 | 0.896 | 0.991 |
| Economy | MSE | 0.061 | 0.570 | 0.056 | 0.013 | 0.015 | 0.069 | 0.015 | 0.012 | 0.019 | 0.153 | 0.145 | 0.039 | 0.016 | 0.020 | 0.045 | 0.016 | 0.015 | 0.022 |
| | MAE | 0.200 | 0.676 | 0.197 | 0.091 | 0.097 | 0.210 | 0.098 | 0.087 | 0.111 | 0.344 | 0.288 | 0.151 | 0.099 | 0.115 | 0.166 | 0.103 | 0.096 | 0.119 |
| Energy | MSE | 0.112 | 0.195 | 0.096 | 0.108 | 0.108 | 0.098 | 0.117 | 0.113 | 0.098 | 0.138 | 0.182 | 0.088 | 0.107 | 0.130 | 0.097 | 0.102 | 0.113 | 0.098 |
| | MAE | 0.240 | 0.347 | 0.215 | 0.234 | 0.239 | 0.225 | 0.240 | 0.246 | 0.225 | 0.260 | 0.311 | 0.202 | 0.229 | 0.262 | 0.224 | 0.221 | 0.243 | 0.221 |
| Environment | MSE | 0.466 | 0.486 | 0.482 | 0.486 | 0.492 | 0.475 | 0.487 | 0.492 | 0.488 | 0.422 | 0.474 | 0.471 | 0.484 | 0.491 | 0.473 | 0.490 | 0.487 | 0.472 |
| | MAE | 0.486 | 0.509 | 0.527 | 0.500 | 0.503 | 0.531 | 0.498 | 0.497 | 0.484 | 0.488 | 0.516 | 0.511 | 0.499 | 0.502 | 0.530 | 0.499 | 0.495 | 0.489 |
| Health | MSE | 1.169 | 1.337 | 1.218 | 2.168 | 2.085 | 1.443 | 1.314 | 1.250 | 1.904 | 1.370 | 1.406 | 1.120 | 1.532 | 1.183 | 1.460 | 1.461 | 1.642 | 1.327 |
| | MAE | 0.676 | 0.767 | 0.754 | 0.842 | 0.815 | 0.809 | 0.770 | 0.745 | 0.810 | 0.833 | 0.790 | 0.732 | 0.761 | 0.717 | 0.821 | 0.734 | 0.745 | 0.742 |
| Socialgood | MSE | 0.914 | 0.756 | 0.872 | 0.863 | 0.954 | 1.030 | 0.902 | 1.086 | 0.982 | 0.957 | 0.785 | 0.887 | 0.863 | 0.936 | 1.013 | 1.019 | 0.972 | 1.327 |
| | MAE | 0.453 | 0.373 | 0.386 | 0.376 | 0.412 | 0.437 | 0.407 | 0.416 | 0.411 | 0.460 | 0.443 | 0.393 | 0.376 | 0.413 | 0.434 | 0.439 | 0.389 | 0.498 |
| Traffic | MSE | 0.182 | 0.133 | 0.205 | 0.147 | 0.187 | 0.153 | 0.159 | 0.176 | 0.152 | 0.180 | 0.136 | 0.204 | 0.147 | 0.189 | 0.152 | 0.162 | 0.194 | 0.167 |
| | MAE | 0.259 | 0.227 | 0.230 | 0.206 | 0.222 | 0.200 | 0.229 | 0.216 | 0.205 | 0.244 | 0.239 | 0.231 | 0.206 | 0.225 | 0.204 | 0.237 | 0.223 | 0.214 |
| Fashion | MSE | 0.485 | 0.454 | 0.629 | 0.525 | 0.555 | 0.474 | 0.513 | 0.592 | 0.427 | 0.482 | 0.516 | 0.618 | 0.524 | 0.558 | 0.474 | 0.513 | 0.592 | 0.471 |
| | MAE | 0.468 | 0.484 | 0.617 | 0.523 | 0.554 | 0.515 | 0.512 | 0.587 | 0.408 | 0.472 | 0.561 | 0.607 | 0.523 | 0.556 | 0.516 | 0.511 | 0.588 | 0.447 |
| Weather | MSE | 0.184 | 0.171 | 0.184 | 0.175 | 0.174 | 0.169 | 0.172 | 0.173 | 0.173 | 0.248 | 0.167 | 0.178 | 0.178 | 0.176 | 0.169 | 0.172 | 0.173 | 0.170 |
| | MAE | 0.325 | 0.315 | 0.326 | 0.315 | 0.314 | 0.314 | 0.313 | 0.313 | 0.314 | 0.379 | 0.309 | 0.320 | 0.319 | 0.317 | 0.314 | 0.315 | 0.313 | 0.310 |
| Medical | MSE | 0.596 | 0.858 | 0.658 | 0.530 | 0.535 | 0.696 | 0.531 | 0.535 | 0.532 | 0.497 | 0.765 | 0.566 | 0.521 | 0.523 | 0.694 | 0.527 | 0.533 | 0.533 |
| | MAE | 0.565 | 0.686 | 0.575 | 0.514 | 0.517 | 0.624 | 0.513 | 0.518 | 0.512 | 0.503 | 0.644 | 0.526 | 0.511 | 0.512 | 0.624 | 0.513 | 0.518 | 0.504 |
| PTF | MSE | 0.110 | 0.108 | 0.194 | 0.100 | 0.104 | 0.134 | 0.110 | 0.102 | 0.118 | 0.089 | 0.105 | 0.142 | 0.099 | 0.101 | 0.120 | 0.088 | 0.105 | 0.090 |
| | MAE | 0.167 | 0.180 | 0.297 | 0.173 | 0.172 | 0.227 | 0.208 | 0.167 | 0.186 | 0.162 | 0.169 | 0.248 | 0.174 | 0.170 | 0.215 | 0.189 | 0.172 | 0.174 |
| MSPG | MSE | 0.365 | 0.380 | 0.353 | 0.324 | 0.415 | 0.415 | 0.355 | 0.390 | 0.454 | 0.300 | 0.397 | 0.470 | 0.363 | 0.466 | 0.408 | 0.353 | 0.376 | 0.424 |
| | MAE | 0.217 | 0.218 | 0.262 | 0.229 | 0.232 | 0.227 | 0.248 | 0.228 | 0.240 | 0.198 | 0.217 | 0.416 | 0.251 | 0.258 | 0.226 | 0.244 | 0.224 | 0.234 |
| LEU | MSE | 0.522 | 0.607 | 0.665 | 0.504 | 0.508 | 0.497 | 0.491 | 0.547 | 0.547 | 0.513 | 0.578 | 0.658 | 0.504 | 0.508 | 0.495 | 0.490 | 0.555 | 0.555 |
| | MAE | 0.350 | 0.404 | 0.444 | 0.381 | 0.376 | 0.382 | 0.372 | 0.385 | 0.337 | 0.339 | 0.405 | 0.448 | 0.386 | 0.379 | 0.382 | 0.371 | 0.388 | 0.345 |
| MTFinance | MSE | 0.002 | 0.107 | 0.003 | 0.003 | 0.003 | 0.002 | 0.003 | 0.005 | 0.002 | 0.004 | 0.033 | 0.003 | 0.016 | 0.031 | 0.002 | 0.005 | 0.002 | 0.003 |
| | MAE | 0.018 | 0.056 | 0.024 | 0.014 | 0.014 | 0.014 | 0.015 | 0.019 | 0.013 | 0.023 | 0.043 | 0.033 | 0.033 | 0.081 | 0.015 | 0.020 | 0.014 | 0.013 |
| MTWeather | MSE | 0.209 | 0.218 | 0.224 | 0.210 | 0.199 | 0.203 | 0.203 | 0.204 | 0.220 | 0.220 | 0.209 | 0.231 | 0.207 | 0.197 | 0.202 | 0.187 | 0.205 | 0.243 |
| | MAE | 0.347 | 0.352 | 0.360 | 0.346 | 0.337 | 0.338 | 0.339 | 0.339 | 0.351 | 0.351 | 0.344 | 0.366 | 0.343 | 0.336 | 0.337 | 0.325 | 0.338 | 0.355 |

We also evaluate both unimodal and aligning-based MMTS using different time series models (Chronos, Informer, FEDformer, PatchTST, iTransformer, DLinear), as well as additional aligning-

based models (Time-LLM, TimeCMA, LeRet) as detailed in Table 11. The corresponding aggregated results across datasets are visualized in Figure 6 of the main text.

## D.2 QWEN 1.5B AS TEXT ENCODER FOR ALIGNING-BASED PARADIGM

In this section, we conduct additional experiments using Qwen 1.5B as the text encoder within our aligning-based paradigm. We explore different configuration options by varying both the layer from which embeddings are extracted and the pooling strategy used to generate fixed-length representations.

Table 12 presents performance across different configurations. We extract embeddings from either the last layer (layer -1) or an intermediate layer (layer 2), and apply either average pooling (Avg) across all tokens or use only the last token (Last) as the representation. For comparison, we also include results from BERT with its standard configuration (last layer with average pooling).

Table 12: Performance comparison of different text encoder configurations using Qwen 1.5B and BERT-Base across seven time series domains. Each domain reports both MSE and MAE metrics.

| Backbone | Extracted Layer | Pooling | Medical | | Environment | | LEU | | PTF | | MSPG | | Agriculture | | Economy | |
|---|---|---|---|---|---|---|---|---|---|---|---|---|---|---|---|---|
| | | | MSE | MAE | MSE | MAE | MSE | MAE | MSE | MAE | MSE | MAE | MSE | MAE | MSE | MAE |
| Qwen 1.5B | -1 | Avg | 0.845 | 0.577 | 0.464 | 0.506 | 0.532 | 0.351 | 0.101 | 0.159 | 0.429 | 0.237 | 0.130 | 0.256 | 0.125 | 0.303 |
| Qwen 1.5B | 2 | Avg | 0.634 | 0.588 | 0.453 | 0.489 | 0.575 | 0.367 | 0.107 | 0.162 | 0.364 | 0.219 | 0.139 | 0.249 | 0.385 | 0.563 |
| Qwen 1.5B | -1 | Last | 0.752 | 0.651 | 0.468 | 0.513 | 0.595 | 0.367 | 0.107 | 0.161 | 0.439 | 0.234 | 0.136 | 0.259 | 0.180 | 0.311 |
| Qwen 1.5B | 2 | Last | 0.659 | 0.606 | 0.462 | 0.516 | 0.563 | 0.364 | 0.104 | 0.162 | 0.420 | 0.230 | 0.149 | 0.257 | 0.087 | 0.251 |
| BERT-Base | -1 | Avg | 0.497 | 0.503 | 0.422 | 0.488 | 0.513 | 0.339 | 0.089 | 0.162 | 0.300 | 0.198 | 0.167 | 0.299 | 0.211 | 0.414 |

Our results reveal several interesting observations. Despite being a much larger model (1.5B parameters), Qwen 1.5B (Qwen et al., 2025) consistently underperforms compared to BERT-Base (Devlin et al., 2019a) (110M parameters) across most domains when used as a text encoder in the aligning-based paradigm. For Qwen 1.5B, performance varies significantly based on both the layer selection and pooling strategy, with the optimal choice varying across domains. Notably, in the smaller datasets (Agriculture and Economy), the performance variations between different Qwen configurations are more pronounced. For instance, in the Economy domain, MSE ranges from 0.087 to 0.385 depending on the configuration, highlighting the sensitivity of model performance to architectural choices on smaller datasets.

These findings align with our discussions in the main paper regarding the different roles of text models in prompting versus aligning-based paradigms. In the aligning-based approach, smaller models like BERT-Base can be effective as text encoders since the primary pattern recognition burden is handled by dedicated time series models. The larger Qwen 1.5B, while potentially more powerful for understanding complex text in prompting scenarios, does not translate this advantage when used purely for feature extraction in the aligning-based framework.

## D.3 CROSS-DOMAIN SETTING FOR ALIGNING-BASED METHODS

We further conducted cross-domain evaluations for aligning-based methods versus unimodal baselines using Chronos model as an example. Specifically, we trained and validated models on four domains (Agriculture, Climate, Economy, Socialgood) and evaluated on the held-out Traffic domain. The results, summarized in Table 13, show consistent trends with our main findings that MMTS does not consistently outperform the strongest unimodal models under cross-domain conditions.

Table 13: Cross-domain evaluation for aligning-based methods.

| Paradigm | MSE | MAE |
|---|---|---|
| Unimodal | 0.373 | 0.457 |
| Aligning | 0.353 | 0.472 |

## D.4 ULTRA-LONG TIME SERIES

Table 14: Performance of aligning-based model with various down-sampling ratios on context length.

| Category | Ratio | MAE | MSE |
|----------|-------|-----|-----|
| PTF | 1 | 0.048 | 0.206 |
| | 0.5 | 0.023 | 0.212 |
| | 0.25 | 0.027 | 0.263 |
| MSPG | 1 | 0.098 | 0.215 |
| | 0.5 | 0.105 | 0.237 |
| | 0.25 | 0.088 | 0.218 |
| LEU | 1 | 0.011 | 0.038 |
| | 0.5 | 0.005 | 0.004 |
| | 0.25 | 0.002 | 0.009 |
| MTFinance | 1 | -0.376 | -1.066 |
| | 0.5 | -0.035 | -0.015 |
| | 0.25 | -0.168 | -0.228 |
| MTWeather | 1 | -0.020 | -0.051 |
| | 0.5 | -0.034 | -0.077 |
| | 0.25 | -0.013 | -0.036 |

We additionally run experiments on an extra-long time series dataset PixelRec Zhou et al. (2025). This dataset captures short video behavior with user interactions aggregated into daily view series. Each time series has context length of 365 and future length of 943. We use Chronos model as an example, and its unimodal and aligning-based counterparts perform comparably on this extremely-long dataset as well, consistent with our main findings.

Table 15: Performance of unimodal and aligning-based models on ultra-long time series.

| Metric | Unimodal | Aligning |
|--------|----------|----------|
| MSE | 0.281 | 0.280 |
| MAE | 0.163 | 0.162 |

# E ADDITIONAL RESULTS FOR PROMPTING-BASED METHODS

## E.1 REASONING LLMs

Table 16: Comparison of deepseek distilled Qwen 1.5B and deepseek distilled Qwen 7B on selected datasets and metrics (MAE, MSE).

| Dataset | Metric | deepseek distilled Qwen 1.5B | deepseek distilled Qwen 7B |
|---|---|---|---|
| Climate | MAE | 1.109 | 0.969 |
| | MSE | 2.051 | 1.437 |
| Weather | MAE | 0.453 | 0.406 |
| | MSE | 0.701 | 0.344 |
| Environment | MAE | 0.755 | 0.645 |
| | MSE | 2.776 | 0.843 |
| Medical | MAE | 1.884 | 0.595 |
| | MSE | 1.909 | 0.742 |
| PTF | MAE | 1.017 | 0.609 |
| | MSE | 1.992 | 0.667 |
| MSPG | MAE | 11.573 | 0.597 |
| | MSE | 23.274 | 4.914 |
| LEU | MAE | 4.231 | 0.529 |
| | MSE | 1.103 | 1.137 |
| MTFinance | MAE | 1.630 | 1.570 |
| | MSE | 0.600 | 0.700 |
| MTWeather | MAE | 2.683 | 1.722 |
| | MSE | 1.827 | 1.362 |

For the reasoning experiment as discussed in Section 4.2, we utilize distilled versions of the DeepSeek-R1 (DeepSeek-AI, 2025) architecture, specifically DeepSeek-R1-Distill-Qwen-1.5B and DeepSeek-R1-Distill-Qwen-7B. These models represent a fusion of the reasoning-centric training of DeepSeek-R1 with the architectural foundations of the Qwen series of large language models. By employing a distillation process, these models aim to inherit the strong reasoning capabilities of DeepSeek-R1 while benefiting from the efficient architecture and pre-training of the Qwen models. The "1.5B" and "7B" suffixes indicate the approximate number of parameters in each model, offering different trade-offs between model capacity and computational efficiency. These distilled models were accessed via **Huggingface** at deepseek-ai/deepseek-r1-distill-qwen-1.5b and deepseek-ai/deepseek-r1-distill-qwen-7b, respectively. Their architecture and training likely incorporate techniques from both DeepSeek-R1's reinforcement learning approach for reasoning and Qwen's extensive pre-training on diverse datasets. One response of DeepSeek-R1-Distill-Qwen-7B can be found as below:

---

**One example reasoning process from DeepSeek-R1-Distill-Qwen-7B.**

```
Okay, I need to help the user with their time series
forecasting task. Let's see what they've provided. First, the
context is empty, so I don't  have any specific background
information. The historical data has eight points, from index
0 to 7. The values are:

(0, 264778)
(1, 242600)
(2, 225644)
(3, 268343)
(4, 272864)
(5, 281264)
(6, 280290)
(7, 288566)
I need to predict the next six timestamps, which would be
```

```
indices 8 through 13.
Looking at the data, there's a dip from index 0 to 2, then it
starts increasing again. From index 3 onwards, the values are
consistently rising, with a slight dip at index 6. The trend
seems upward, but there's some variability.

Since there's no context about seasonality or specific models
to use, I'll consider a simple method like the average of the
last few points or maybe a linear trend. Alternatively,
exponential smoothing could be an option, but without knowing
the parameters, it's tricky.

Another approach is to look at the differences between
consecutive points to see if there's a pattern. Let me
calculate the differences:

From 0 to 1: 242600 - 264778 = -22178
1 to 2: 225644 - 242600 = -16956
2 to 3: 268343 - 225644 = +42699
3 to 4: 272864 - 268343 = +4521
4 to 5: 281264 - 272864 = +8400
5 to 6: 280290 - 281264 = -974
6 to 7: 288566 - 280290 = +8276

The differences are quite variable. There's a big jump from 2
to 3, then smaller increases and a slight dip. It's hard to
see a clear trend here.Maybe the recent values are more
stable, so I could take the average of the last few points.

Looking at the last three points (indices 5,6,7): 281264,
280290, 288566. The average is (281264 + 280290 + 288566)/3
$\approx$ 283373.33. So, maybe the next value could be around
there, and perhaps a slight increase or decrease.

Alternatively, considering the last value is 288566, which is
higher than the previous, maybe the trend is upward. So, I
could add a small increment each time.

But without more data or context, it's challenging. Maybe a
simple moving average or just taking the last value as a base
and adding a small percentage.

Wait, the user didn't specify the method, so perhaps I should
use a simple approach like taking the last value and adding a
small increase each time. Let's say the last value is 288566,
and I'll assume a 2% increase each period. That might be too
high, though. Alternatively, looking at the increase from 6 to
7: 288566 - 280290 = 8276. Maybe I'll use that as the
increment.

So, index 8: 288566 + 8276 = 296842
Index 9: 296842 + 8276 = 305118
Index 10: 305118 + 8276 = 313394
Index 11: 313394 + 8276 = 321670
Index 12: 321670 + 8276 = 329946
Index 13: 329946 + 8276 = 338222
```

But this assumes a constant increase, which might not be accurate. Alternatively, I could consider the average increase over the last few periods. The increases from 5 to 7 are +8400, −974, +8276. The average increase is (8400 − 974 + 8276)/3 $\approx$ (8400 + 8276 − 974)/3 $\approx$ (16676 − 974)/3 $\approx$ 15702/3 $\approx$ 5234. So, maybe add around 5000 each time.

But the last increase was 8276, which is higher than the average. Maybe I'll take a weighted average, giving more weight to the recent increase.

Alternatively, perhaps a linear regression on the last few points. Let me try that.

Using the last three points (indices 5,6,7):

x: 5,6,7
y:281264,280290,288566

Calculate the slope:

Sum of x: 5+6+7=18
Sum of y:281264+280290+288566=850120
Sum of xy:5281264 +6280290 +7288566

Calculate each term:

5281264 = 1,406,320
6280,290 = 1,681,740
7288,566 = 2,019,962
Sum of xy: 1,406,320 + 1,681,740 = 3,088,060 + 2,019,962 = 5,108,022

n=3
Slope (b) = (nsum(xy) − sum(x)sum(y)) / (nsum(x²) − (sum(x))²)

sum(x²) = 25 + 36 + 49 = 110

So numerator: 35,108,022 − 18850,120 = 15,324,066 − 15,302,160 = 21,906

Denominator: 3110 − 18² = 330 − 324 = 6

So slope b = 21,906 /6 $\approx$ 3,651

Intercept a = (sum(y) − bsum(x))/n = (850,120 − 3,65118)/3

3,65118 = 65,718

850,120 − 65,718 = 784,402

a = 784,402 /3 $\approx$ 261,467.33

So the linear regression equation is y = 261,467.33 + 3,651x

Now, to predict for x=8:

```
y = 261,467.33 + 3,6518 $\approx$ 261,467.33 + 29,208 $\approx$

290,675.33

Similarly for x=9: 261,467.33 + 3,6519 $\approx$ 261,467.33 +
32,859
$\approx$ 294,326.33

x=10: 261,467.33 + 3,65110 $\approx$ 261,467.33 + 36,510
$\approx$ 297,977.33

x=11: 261,467.33 + 3,65111 $\approx$ 261,467.33 + 40,161
$\approx$ 301,628.33

x=12: 261,467.33 + 3,65112 $\approx$ 261,467.33 + 43,812
$\approx$ 305,279.33

x=13: 261,467.33 + 3,651*13 $\approx$ 261,467.33 + 47,463
$\approx$ 308,930.33

These are the predicted values using linear regression on the
last three points. They show a steady increase, which makes
sense given the trend in the data.

Alternatively, I could use a different method like moving
average or exponential smoothing, but without more context,
linear regression on the recent data seems reasonable.

So, I'll go with these predictions.
</think>

<forecast>
(8, 290675)
(9, 294326)
(10, 297977)
(11, 301628)
(12, 305279)
(13, 308930)
</forecast>
```

## E.2 IN-CONTEXT LEARNING AND CHAIN OF THOUGHT

We conducted three additional experiments for prompting-based methods:

1. **LLM Process (LLMP)** (Requeima et al., 2024). We use Mistral $8\times7B$ for LLMP, as it shows slightly better performance than other models (Requeima et al., 2024).

2. **In-Context Learning (ICL).** For each dataset, we randomly select three in-context examples from the training set and evaluate on Qwen-32B and Claude 3.5.

> **ICL prompt inserted at the beginning of the optimized prompt**
>
> ```
> # Context-Informed Time Series Forecasting
>
> You will be shown several examples of time series forecasting,
> ```

```
followed by a new case to forecast.

## Examples:
{examples_text}

## Now, forecast for this new case:

...
```

3. **Chain-of-Thought (CoT).** We prompt Qwen-32B and Claude 3.5 to reason step by step, following Wei et al. (2022).

**CoT prompt inserted at the end of the optimized prompt**

```
Please solve this problem by:
1. First, identify what we know
2. Then, determine what we need to find
3. Next, work through the solution step by step
4. Finally, provide the answer
```

Based on the newly added methods, our conclusion remains the same: (1) Aligning-based models tend to outperform prompting-based methods, achieving the lowest MSE and MAE. (2) For prompting-based methods, we found direct prompting performs better than LLMP, which is consistent with Williams et al. (2025).

Table 17: Additional prompting-based methods.

| Dataset | ICL Claude 3.5 | | ICL Qwen 32B | | CoT Claude 3.5 | | CoT Qwen 32B | | LLMP Mistral 8x7B | |
|---|---|---|---|---|---|---|---|---|---|---|
| Metrics | MSE | MAE | MSE | MAE | MSE | MAE | MSE | MAE | MSE | MAE |
| Agriculture | 0.095 | 0.185 | 0.122 | 0.224 | 0.098 | 0.222 | 0.107 | 0.219 | 0.541 | 0.683 |
| Climate | 0.882 | 0.963 | 0.911 | 0.675 | 1.589 | 0.983 | 1.425 | 0.962 | 1.909 | 2.297 |
| Economy | 0.014 | 0.074 | 0.030 | 0.133 | 0.036 | 0.152 | 0.038 | 0.155 | 0.004 | 0.681 |
| Fashion | 0.535 | 0.472 | 0.570 | 0.502 | 0.555 | 0.520 | 0.542 | 0.489 | 1.417 | 0.951 |
| Energy | 0.121 | 0.227 | 0.097 | 0.209 | 0.124 | 0.244 | 0.124 | 0.231 | 0.390 | 0.833 |
| Environment | 0.725 | 0.599 | 0.649 | 0.560 | 0.693 | 0.590 | 1.011 | 0.668 | 1.099 | 1.434 |
| Health | 1.849 | 1.276 | 1.118 | 0.885 | 3.858 | 1.234 | 2.655 | 0.969 | 0.271 | 0.608 |
| Socialgood | 0.840 | 0.346 | 0.750 | 0.402 | 0.740 | 0.405 | 0.781 | 0.417 | 0.469 | 0.637 |
| Traffic | 0.185 | 0.205 | 0.264 | 0.345 | 0.204 | 0.259 | 0.279 | 0.391 | 2.863 | 1.262 |
| Weather | 0.262 | 0.376 | 0.242 | 0.368 | 0.253 | 0.369 | 0.268 | 0.376 | 0.573 | 0.787 |
| Medical | 0.582 | 0.538 | 0.603 | 0.543 | 0.706 | 0.577 | 0.678 | 0.559 | 0.967 | 1.133 |
| PTF | 0.220 | 0.306 | 0.475 | 0.429 | 0.321 | 0.341 | 0.588 | 0.560 | 0.969 | 1.216 |
| MSPG | 1.998 | 0.407 | 1.637 | 0.488 | 1.436 | 0.585 | 0.932 | 0.458 | 0.403 | 0.354 |
| LEU | 1.928 | 0.523 | 1.464 | 0.682 | 1.636 | 0.508 | 1.464 | 0.682 | 0.617 | 0.479 |
| MTFinance | 0.001 | 0.017 | 0.003 | 0.014 | 0.003 | 0.015 | 0.004 | 0.017 | 0.003 | 0.013 |
| MTWeather | 2.090 | 0.903 | 0.611 | 0.827 | 1.525 | 0.830 | 0.463 | 0.494 | 0.953 | 0.603 |
| **Average** | 0.731 | 0.464 | 0.638 | 0.460 | 0.875 | 0.499 | 0.763 | 0.499 | 0.892 | 0.954 |

### E.3 FINETUNING LLM

Most MMTS forecasting studies rely on direct prompting, yet recent work especially on time series reasoning performs finetuning (Merrill et al., 2024). Motivated by this, we fine-tuned Mistral 7B-Instruct v0.2[8] on the Time-Series Reasoning corpus[9], casting each sample as a forecasting problem with a horizon of 30 time steps. We concatenate the historical values, the dataset's `Description` field (used as textual context), and the target sequence into a single stream, tokenize with the native

---

[8]https://huggingface.co/mistralai/Mistral-7B-Instruct-v0.2
[9]https://huggingface.co/datasets/mikeam/time-series-reasoning

Mistral tokenizer, and truncate or pad to 5,120 tokens. Training proceeds for 20,000 steps on one NVIDIA A100-40GB GPU in bfloat16, using full-parameter AdamW (learning rate $2.4 \times 10^{-4}$, weight decay 0.1) with a cosine decay schedule and a linear warm-up over the first 5 % of updates; the global batch size is 4, checkpoints are written every 1,000 steps, and the best model by validation loss is kept after roughly 16 hours of wall time. All runs fix the random seed to 42.

Fine-tuning reduces error relative to direct prompting, yet a substantial gap to strong unimodal baselines remains: our fine-tuned Mistral reaches a (lower-is-better) Mean Absolute Scaled Error (MASE) of **1.702**, whereas the dedicated time-series model `Chronos` attains a MASE of **1.492**. This outcome is consistent with (Merrill et al., 2024), who notes that LLMs "struggle to learn etiological relationships between time series and text" even after fine-tuning.

### E.4    THE IMPACT OF CONTEXT LENGTH ON DIRECT PROMPTING

We evaluate direct prompting under different context budgets by truncating the available textual history to full, half, and quarter length across five categories (PTF, MSPG, LEU, MTFinance, MTWeather) and tracking changes in forecasting error (MAE and MSE). Although giving the model the full context generally yields the strongest gains, it exhibits large dispersion across different datasets. Quarter-length context often exhibits comparable performance compared to halving the context. Therefore, we do not observe an obvious monotonous trend from 0.25 to 1. This mirrors what we see with alignment-based multimodal methods.

### E.5    VISION LANGUAGE MODELS

We also added an experiment converting time series into images (TS-as-image) and used a vision-language model Claude 3.7 to forecast in Table 18. We compared three scenarios: TS-as-image+text+TS, text+TS and TS-as-image+TS, which perform comparably with each other. These results are consistent with our analysis on the text modality, which also point to the importance of complementary, future-predictive information. Our conclusion also remains that numerical reasoning for forecasting remains a significant challenge for current LLMs, regardless of size or architecture.

Table 18: Performance of vision language model.

| Dataset | TS-as-image + Text + TS | | Text + TS | | TS-as-image + TS | |
|---|---|---|---|---|---|---|
| Metrics | MAE | MSE | MAE | MSE | MAE | MSE |
| Agriculture | 0.236 | 0.122 | 0.242 | 0.132 | 0.246 | 0.133 |
| Climate | 0.956 | 1.464 | 0.984 | 1.492 | 0.967 | 1.487 |
| Economy | 0.164 | 0.043 | 0.132 | 0.027 | 0.158 | 0.039 |
| Energy | 0.228 | 0.118 | 0.244 | 0.133 | 0.252 | 0.156 |
| Environment | 0.581 | 0.646 | 0.590 | 0.694 | 0.584 | 0.651 |
| Health | 1.180 | 5.276 | 1.140 | 2.966 | 1.333 | 5.949 |
| Socialgood | 0.450 | 0.897 | 0.419 | 0.734 | 0.461 | 0.937 |
| Traffic | 0.382 | 0.318 | 0.238 | 0.168 | 0.408 | 0.358 |
| Fashion | 0.502 | 0.592 | 0.520 | 0.565 | 0.505 | 0.609 |
| Weather | 0.268 | 0.258 | 0.276 | 0.271 | 0.264 | 0.249 |
| Medical | 0.812 | 0.523 | 0.858 | 0.602 | 0.794 | 0.511 |
| PTF | 0.339 | 0.304 | 0.307 | 0.247 | 0.440 | 0.464 |
| MSPG | 0.321 | 0.570 | 0.291 | 0.466 | 0.396 | 0.828 |
| LEU | 0.489 | 0.827 | 0.429 | 0.755 | 0.436 | 0.766 |
| MTFinance | 0.013 | 0.002 | 0.012 | 0.002 | 0.013 | 0.002 |
| MTWeather | 0.482 | 0.432 | 0.496 | 0.441 | 0.507 | 0.449 |
| **Average** | 0.462 | 0.649 | 0.448 | 0.605 | 0.485 | 0.849 |

Table 19: Performance of prompting-based model with respect to various down-sampling ratios on context length.

| Category | Ratio | MAE | MSE |
|---|---|---|---|
| PTF | 1 | 0.211 | 0.214 |
| | 0.5 | 0.057 | 0.095 |
| | 0.25 | 0.001 | 0.060 |
| MSPG | 1 | 0.188 | 0.368 |
| | 0.5 | 0.003 | 0.017 |
| | 0.25 | 0.009 | 0.018 |
| LEU | 1 | -0.002 | 0.058 |
| | 0.5 | -0.069 | -0.052 |
| | 0.25 | -0.002 | 0.310 |
| MTFinance | 1 | 0.076 | 0.500 |
| | 0.5 | 0.021 | 0.583 |
| | 0.25 | -0.262 | -0.220 |
| MTWeather | 1 | 0.012 | 0.127 |
| | 0.5 | -0.023 | -0.099 |
| | 0.25 | -0.076 | -0.193 |

Table 20: MMTS performance for more LLMs, extended results from Table 1. The summarized results can be found at Figure 3.

| Dataset | Metric | Qwen 1.5B | Qwen 7B | LLaMA 3B | LLaMA 8B | LLaMA 70B |
|---|---|---|---|---|---|---|
| Agriculture | MAE | 0.422 | 0.249 | 0.274 | 0.420 | 0.259 |
| | MSE | 0.395 | 0.146 | 0.150 | 0.537 | 0.153 |
| | CRPS | 0.308 | 0.218 | 0.201 | 0.337 | 0.249 |
| | WQL | 0.117 | 0.075 | 0.086 | 0.126 | 0.079 |
| Climate | MAE | 0.982 | 0.976 | 1.164 | 1.862 | 0.992 |
| | MSE | 1.526 | 1.472 | 1.883 | 4.964 | 1.548 |
| | CRPS | 0.683 | 0.834 | 0.917 | 1.631 | 0.912 |
| | WQL | 1.143 | 1.194 | 1.596 | 2.393 | 1.245 |
| Economy | MAE | 0.239 | 0.171 | 0.183 | 0.289 | 0.156 |
| | MSE | 0.451 | 0.044 | 0.053 | 0.134 | 0.037 |
| | CRPS | 0.117 | 0.149 | 0.131 | 0.244 | 0.146 |
| | WQL | 0.059 | 0.056 | 0.060 | 0.097 | 0.053 |
| Energy | MAE | 0.318 | 0.252 | 0.327 | 0.348 | 0.240 |
| | MSE | 0.194 | 0.142 | 0.226 | 0.294 | 0.141 |
| | CRPS | 0.252 | 0.223 | 0.239 | 0.303 | 0.227 |
| | WQL | 0.713 | 0.542 | 0.832 | 0.906 | 0.519 |
| Environment | MAE | 0.633 | 0.582 | 0.942 | 0.856 | 0.725 |
| | MSE | 0.825 | 0.674 | 1.976 | 1.749 | 1.432 |
| | CRPS | 0.434 | 0.511 | 0.457 | 0.652 | 0.503 |
| | WQL | 0.785 | 0.757 | 1.113 | 1.130 | 0.906 |
| Health | MAE | 1.130 | 0.880 | 1.316 | 2.096 | 0.883 |
| | MSE | 2.713 | 1.832 | 3.701 | 2.234 | 2.067 |
| | CRPS | 0.844 | 0.742 | 0.930 | 1.692 | 0.801 |
| | WQL | 1.535 | 1.153 | 1.898 | 2.518 | 1.281 |
| Socialgood | MAE | 0.737 | 0.453 | 0.631 | 0.538 | 0.444 |
| | MSE | 1.483 | 0.809 | 1.394 | 1.069 | 0.623 |
| | CRPS | 0.575 | 0.386 | 0.484 | 0.449 | 0.417 |
| | WQL | 0.935 | 0.546 | 0.866 | 0.575 | 0.634 |
| Traffic | MAE | 0.402 | 0.395 | 0.515 | 0.685 | 0.403 |
| | MSE | 0.293 | 0.295 | 0.436 | 0.767 | 0.309 |
| | CRPS | 0.270 | 0.328 | 0.389 | 0.560 | 0.366 |
| | WQL | 0.253 | 0.276 | 0.331 | 0.473 | 0.287 |
| Fashion | MAE | 0.568 | 0.512 | 0.800 | 0.527 | 0.492 |
| | MSE | 0.717 | 0.600 | 0.774 | 0.610 | 0.563 |
| | CRPS | 0.413 | 0.475 | 0.399 | 0.465 | 0.477 |
| | WQL | 1.079 | 1.049 | 1.179 | 1.049 | 1.015 |
| Weather | MAE | 0.376 | 0.375 | 0.342 | 0.514 | 0.362 |
| | MSE | 0.258 | 0.265 | 0.211 | 0.513 | 0.241 |
| | CRPS | 0.297 | 0.329 | 0.250 | 0.416 | 0.332 |
| | WQL | 0.664 | 0.650 | 0.596 | 0.878 | 0.651 |
| Medical | MAE | 0.635 | 0.556 | 0.787 | 0.744 | 0.581 |
| | MSE | 3.768 | 0.624 | 5.883 | 1.354 | 0.698 |
| | CRPS | 0.424 | 0.493 | 0.443 | 0.601 | 0.546 |
| | WQL | 0.572 | 0.566 | 0.661 | 0.783 | 0.603 |
| PTF | MAE | 0.633 | 0.899 | 0.682 | 0.922 | 0.663 |
| | MSE | 0.770 | 1.353 | 0.912 | 1.495 | 0.953 |
| | CRPS | 0.374 | 0.732 | 0.823 | 0.737 | 0.497 |
| | WQL | 0.728 | 1.141 | 0.354 | 1.187 | 0.815 |
| MSPG | MAE | 0.366 | 0.537 | 0.889 | 0.419 | 0.483 |
| | MSE | 0.892 | 1.257 | 2.566 | 1.203 | 1.267 |
| | CRPS | 0.234 | 0.411 | 0.382 | 1.203 | 0.384 |
| | WQL | 0.475 | 0.876 | 1.247 | 0.608 | 0.845 |
| LEU | MAE | 0.518 | 0.503 | 0.684 | 0.571 | 0.652 |
| | MSE | 0.898 | 0.909 | 1.325 | 1.199 | 2.703 |
| | CRPS | 0.319 | 0.348 | 0.377 | 0.509 | 0.427 |
| | WQL | 0.734 | 0.728 | 1.028 | 0.931 | 0.962 |
| MTFinance | MAE | 0.034 | 0.014 | 0.079 | 0.030 | 0.021 |
| | MSE | 0.244 | 0.002 | 0.420 | 0.011 | 0.004 |
| | CRPS | 0.026 | 0.012 | 0.033 | 0.012 | 0.015 |
| | WQL | 1.075 | 0.085 | 1.045 | 0.058 | 0.086 |
| MTWeather | MAE | 1.200 | 0.588 | 1.236 | 0.926 | 0.662 |
| | MSE | 0.868 | 0.686 | 1.023 | 0.939 | 0.866 |
| | CRPS | 0.355 | 0.368 | 0.446 | 0.418 | 0.415 |
| | WQL | 1.399 | 0.890 | 1.212 | 1.227 | 1.178 |
| Average | MAE | 0.575 | 0.496 | 0.678 | 0.734 | 0.501 |
| | MSE | 1.018 | 0.694 | 1.433 | 1.192 | 0.850 |
| | CRPS | 0.370 | 0.410 | 0.431 | 0.639 | 0.420 |
| | WQL | 0.767 | 0.662 | 0.882 | 0.934 | 0.697 |

Table 21: MAE, MSE, CRPS, and WQL across various LLMs without text context.

| Dataset | Metric | Qwen 1.5B | Qwen 7B | Qwen 32B | LLaMA3.3 70B | GPT-4o (mini) |
|---------|--------|-----------|---------|----------|--------------|---------------|
| Agriculture | MAE | 0.308 | 0.260 | 0.231 | 0.269 | 0.241 |
| | MSE | 0.206 | 0.157 | 0.123 | 0.171 | 0.129 |
| | CRPS | 0.232 | 0.234 | 0.215 | 0.255 | 0.208 |
| | WQL | 0.091 | 0.079 | 0.072 | 0.081 | 0.073 |
| Climate | MAE | 1.013 | 0.933 | 0.986 | 0.984 | 0.967 |
| | MSE | 1.586 | 1.357 | 1.457 | 1.487 | 1.396 |
| | CRPS | 0.685 | 0.758 | 0.810 | 0.885 | 0.748 |
| | WQL | 1.184 | 1.096 | 1.180 | 1.213 | 1.136 |
| Economy | MAE | 0.162 | 0.170 | 0.134 | 0.148 | 0.154 |
| | MSE | 0.041 | 0.045 | 0.028 | 0.033 | 0.035 |
| | CRPS | 0.113 | 0.152 | 0.120 | 0.138 | 0.130 |
| | WQL | 0.051 | 0.057 | 0.045 | 0.050 | 0.050 |
| Energy | MAE | 0.291 | 0.268 | 0.242 | 0.250 | 0.241 |
| | MSE | 0.172 | 0.163 | 0.136 | 0.169 | 0.141 |
| | CRPS | 0.233 | 0.239 | 0.221 | 0.233 | 0.210 |
| | WQL | 0.611 | 0.554 | 0.603 | 0.489 | 0.501 |
| Environment | MAE | 0.667 | 0.606 | 0.700 | 0.732 | 0.558 |
| | MSE | 1.169 | 0.740 | 1.176 | 1.712 | 0.623 |
| | CRPS | 0.429 | 0.487 | 0.458 | 0.488 | 0.392 |
| | WQL | 0.806 | 0.787 | 0.833 | 0.876 | 0.694 |
| Health | MAE | 1.279 | 0.943 | 1.030 | 1.149 | 1.153 |
| | MSE | 14.406 | 2.222 | 2.864 | 8.370 | 4.184 |
| | CRPS | 0.839 | 0.830 | 0.906 | 1.020 | 0.991 |
| | WQL | 1.414 | 1.189 | 1.384 | 1.413 | 1.432 |
| Socialgood | MAE | 0.514 | 0.494 | 0.470 | 0.475 | 0.464 |
| | MSE | 0.974 | 1.002 | 0.889 | 0.858 | 0.872 |
| | CRPS | 0.383 | 0.450 | 0.432 | 0.449 | 0.395 |
| | WQL | 0.645 | 0.605 | 0.606 | 0.632 | 0.546 |
| Traffic | MAE | 0.418 | 0.449 | 0.365 | 0.432 | 0.430 |
| | MSE | 0.300 | 0.341 | 0.274 | 0.329 | 0.319 |
| | CRPS | 0.273 | 0.379 | 0.317 | 0.391 | 0.356 |
| | WQL | 0.268 | 0.309 | 0.260 | 0.305 | 0.295 |
| Fashion | MAE | 0.878 | 0.522 | 0.489 | 0.504 | 0.503 |
| | MSE | 15.826 | 0.619 | 0.572 | 0.594 | 0.593 |
| | CRPS | 0.575 | 0.507 | 0.485 | 0.504 | 0.501 |
| | WQL | 1.476 | 1.105 | 1.025 | 1.078 | 1.073 |
| Weather | MAE | 0.385 | 0.383 | 0.382 | 0.401 | 0.370 |
| | MSE | 0.278 | 0.263 | 0.288 | 0.314 | 0.251 |
| | CRPS | 0.300 | 0.337 | 0.344 | 0.375 | 0.315 |
| | WQL | 0.648 | 0.659 | 0.677 | 0.707 | 0.628 |
| Medical | MAE | 0.583 | 0.578 | 0.577 | 0.600 | 0.545 |
| | MSE | 0.740 | 0.725 | 0.761 | 0.830 | 0.664 |
| | CRPS | 0.447 | 0.501 | 0.519 | 0.562 | 0.459 |
| | WQL | 0.579 | 0.580 | 0.593 | 0.621 | 0.540 |
| PTF | MAE | 0.715 | 0.901 | 0.724 | 0.762 | 0.619 |
| | MSE | 0.933 | 1.401 | 1.059 | 1.157 | 0.667 |
| | CRPS | 0.446 | 0.701 | 0.572 | 0.593 | 0.446 |
| | WQL | 0.854 | 1.079 | 0.890 | 0.975 | 0.755 |
| MSPG | MAE | 0.311 | 0.859 | 0.494 | 0.419 | 0.498 |
| | MSE | 0.660 | 3.421 | 1.594 | 1.183 | 1.654 |
| | CRPS | 0.216 | 0.713 | 0.483 | 0.417 | 0.470 |
| | WQL | 0.439 | 0.892 | 0.753 | 0.728 | 0.719 |
| LEU | MAE | 0.487 | 0.512 | 0.451 | 0.527 | 0.506 |
| | MSE | 0.860 | 0.962 | 0.870 | 1.485 | 0.963 |
| | CRPS | 0.320 | 0.382 | 0.361 | 0.354 | 0.378 |
| | WQL | 0.701 | 0.760 | 0.674 | 0.750 | 0.731 |
| MTFinance | MAE | 0.037 | 0.032 | 0.014 | 0.015 | 0.012 |
| | MSE | 0.018 | 0.007 | 0.003 | 0.002 | 0.002 |
| | CRPS | 0.031 | 0.027 | 0.014 | 0.014 | 0.012 |
| | WQL | 0.055 | 0.483 | 0.062 | 0.044 | 0.059 |
| MTWeather | MAE | 0.614 | 0.604 | 0.631 | 0.698 | 0.546 |
| | MSE | 0.731 | 0.743 | 0.809 | 1.049 | 0.520 |
| | CRPS | 0.463 | 0.322 | 0.527 | 0.566 | 0.456 |
| | WQL | 1.032 | 0.986 | 1.042 | 1.246 | 0.854 |
| Average | MAE | 0.541 | 0.532 | 0.495 | 0.523 | 0.488 |
| | MSE | 2.431 | 0.885 | 0.806 | 1.234 | 0.813 |
| | CRPS | 0.374 | 0.439 | 0.424 | 0.453 | 0.404 |
| | WQL | 0.678 | 0.701 | 0.669 | 0.700 | 0.630 |

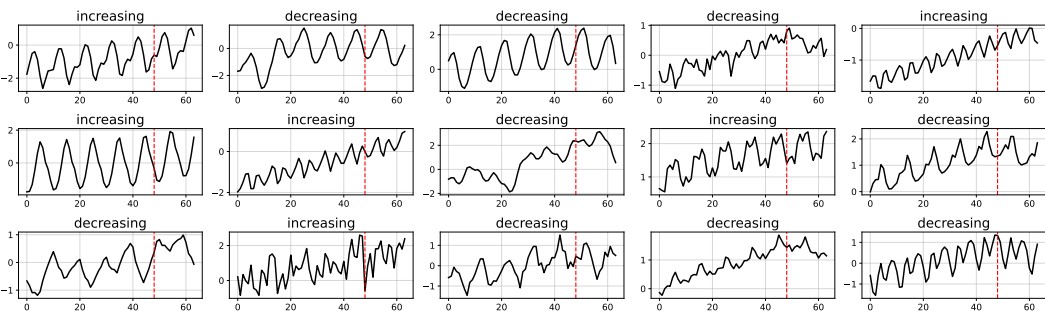

Figure 12: Varying trend where text contains unique information.

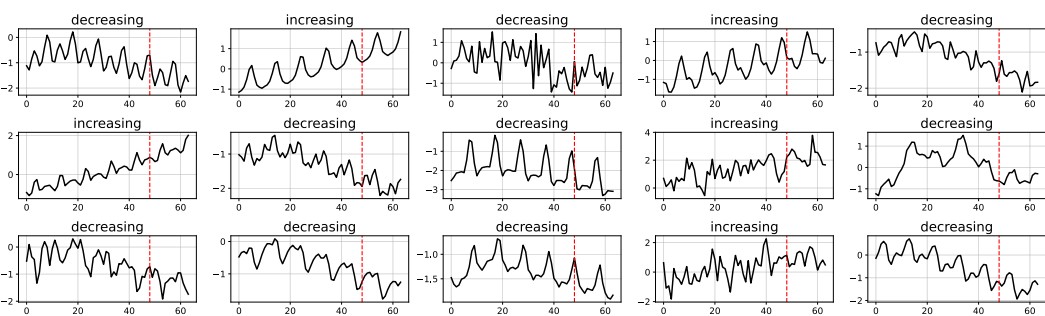

Figure 13: Varying trend where text contains redundant information.

# F  SYNTHETIC TIME SERIES GENERATION

As described in Section 4.6, we construct synthetic MMTS datasets by explicitly controlling the relationship between text and time series. All time series are generated using Gaussian Processes (GPs). We consider three distinct scenarios (Trend, Seasonality, and Spikes), to study how the text encodes information that is either unique (only available through text) or redundant (already present in the time series).

**Trend.** Example time series–text pairs are shown in Figures 12 and 13. Figure 12 illustrates unique information, where the trend direction increases or decreases exactly at the forecasting point and is only indicated in the text. In contrast, Figure 13 demonstrates redundant information, where the trend direction already changes within the observed time series context.

**Seasonality.** Figures 14 and 15 show time series–text examples for the seasonality setting. In Figure 14, the periodicity increases or decreases at the forecasting point, providing unique information that is captured by the text. Figure 15, on the other hand, presents redundant information, where the period shifts within the input context. For simplicity, the time series generated from GP are stationary.

**Spikes.** We present examples in Figures 16 and 17. In Figure 16, the text reveals unique information about whether a spike will occur at the next seasonal peak, which is not visible from the input context alone. Conversely, Figure 17 depicts redundant information, where future spikes can be inferred from earlier spike patterns already observed in the context. For simplicity, the time series generated from GP are stationary.

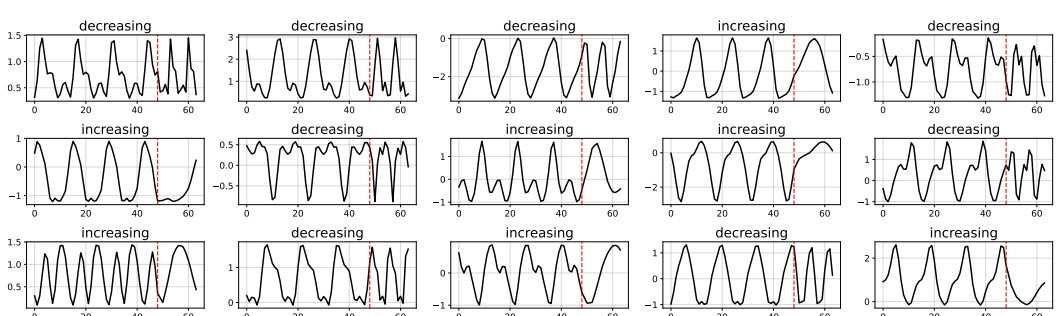

Figure 14: Varying frequency where text contains unique information.

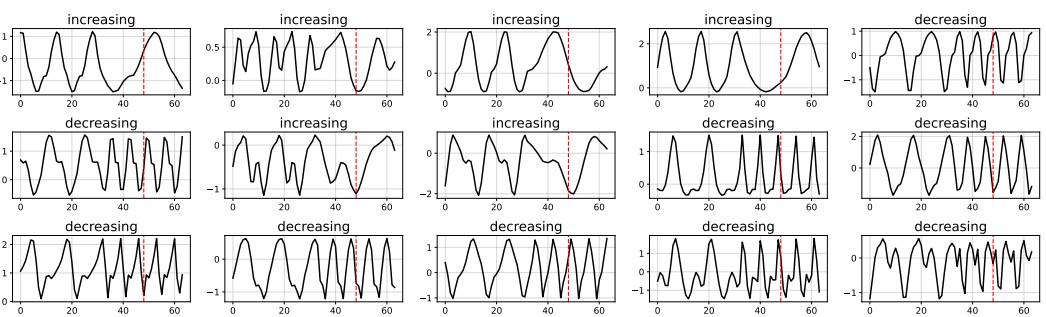

Figure 15: Varying frequency where text contains redundant information.

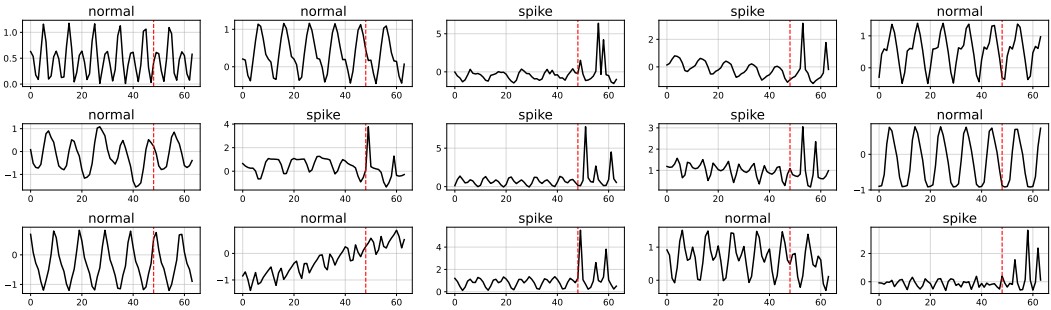

Figure 16: Varying spike where text contains unique information.

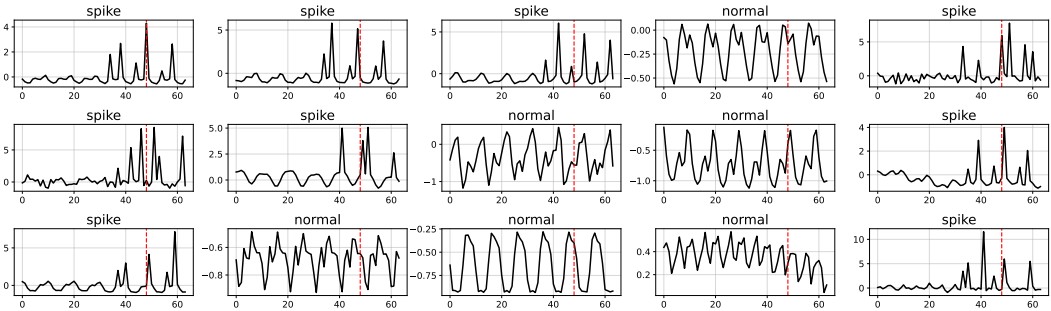

Figure 17: Varying spike where text contains redundant information.



Figure 18: Signal to noise ratio with respect to MMTS performance. Aggregated results of these datasets except Fashion (which only has a one-step history and therefore lacks history descriptions), are shown in Figure 11.

## G  SYNTHETIC TEXT GENERATION

We provide example visualizations of LLM-generated time series history, future descriptions and events in Figure 19, 20, 21, 22, 23, 24, 25, 26, 27, 28. The first row of each figure is the LLM descriptions on history time series (denoted as "History" in Figure 11 and Figure 18); the second row is the LLM descriptions on future time series (denoted as "LLM Desc" in Figure 11 and Figure 18); and the last row is the LLM-generated events that mimic real-world events that could plausibly explain the observed time series patterns (denoted as "LLM Event" in Figure 11 and Figure 18).

### G.1  TIME SERIES DESCRIPTION USING LLM

In Section 4.6, to generate natural language descriptions of the time series data (denoted as "LLM Desc" for future time series and "History" for history time series in Figure 11 and Figure 18), we employed the Claude 3.5 Sonnet model accessed via Amazon Bedrock. The LLM was prompted with a specific instruction set designed to elicit concise summaries of the key patterns and trends within each univariate time series, while filtering out minor fluctuations. The prompt used was as follows:

---

**Prompt to generation synthetic description of the time series**

```
You are a data analyst assistant. Given a univariate time
series, generate a natural language description that briefly
summarizes the key patterns, trends, and characteristics of
the time series. Focus on the overall features and omit the
obvious noise (small fluctations).

Focus on describing:
- The overall trend (increasing, decreasing, stable,
nonlinear, etc.)
- Significant changes (sudden jumps, drops, or trend reversals)
- Anything notable about the shape of the curve
- Omit the obvious noise (small fluctations)

<time_series>
{time_series}
</time_series>

Please write a concise but informative summary of the time
series in natural language. Your output must be wrapped
strictly within <summary> and </summary> tags. Do not include
any text outside these tags. Limit the answer wthin 50 words.
Do not mention the exact numbers.
```

---

The LLM was provided with the time series data within the <time_series> tags, and the generated summary was expected to be enclosed within <summary> and </summary> tags, adhering to the

specified length and content constraints. This approach allowed us to obtain consistent and focused textual descriptions of the temporal patterns for further analysis or interpretation.

## G.2 REAL EVENTS THAT MIMIC TEXT GENERATION

In addition to generating general summaries, we also prompted the LLM to create text that mimics real-world events that could plausibly explain the observed time series patterns, while staying coherent with the domain of the data (denoted as "LLM Event" in Figure 11 and Figure 18). For this task, we again used the Claude 3.5 Sonnet model via Amazon Bedrock with the following prompt:

---

**Prompt to generation synthetic descriptions of the time series that mimic real events**

```
You are a data analyst assistant who explains time series
behaviors by imagining real-world events that could cause
them. Your explanation must be plausible and align with the
dataset's domain.

{domain_hint}

The generated text need to coherent with the change of the
time series:
- The overall trend (increasing, decreasing, stable,
nonlinear, etc.)
- Any significant changes (sudden jumps, drops, or trend
reversals)
- Notable shape features (e.g., plateau, dip, spike)
- Ignore minor fluctuations or noise

Do not mention specific numeric values. Your response must be
wrapped strictly within <summary> and </summary> tags. Keep
the explanation under 50 words and avoid any text outside the
tags. Below is a univariate time series. Based on the trend,
change points, and curve shape, imagine a plausible real-world
event that could explain the pattern. Stay coherent with the
domain.

<time_series>
{time_series}
</time_series>

Respond with only a natural-sounding explanation enclosed
strictly within <summary> and </summary> tags. No numbers. No
extra text. Limit to 100 words.
```

---

To ensure that LLM generates plausible real-world event explanations, we provided domain-specific hints based on the filename of the time series data. Table 22 outlines the mapping between filename keywords and the corresponding domain hints used in the prompt. Each datasets will get mapped to the most plausible domains based on the dataset name. For example, the weather dataset and MTweather will have the same domain hint.

These domain hints were inserted into the prompt's {domain_hint} placeholder to guide the LLM in generating contextually relevant and plausible explanations for the observed time series behaviors.

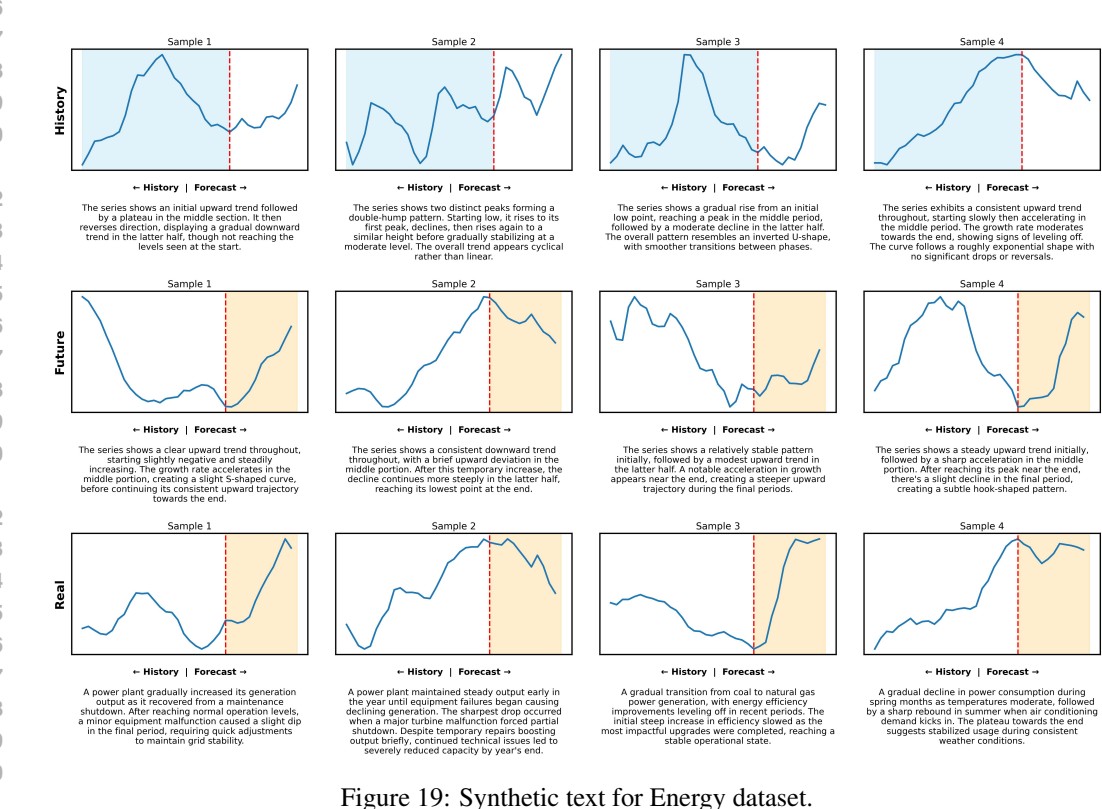

Figure 19: Synthetic text for Energy dataset.

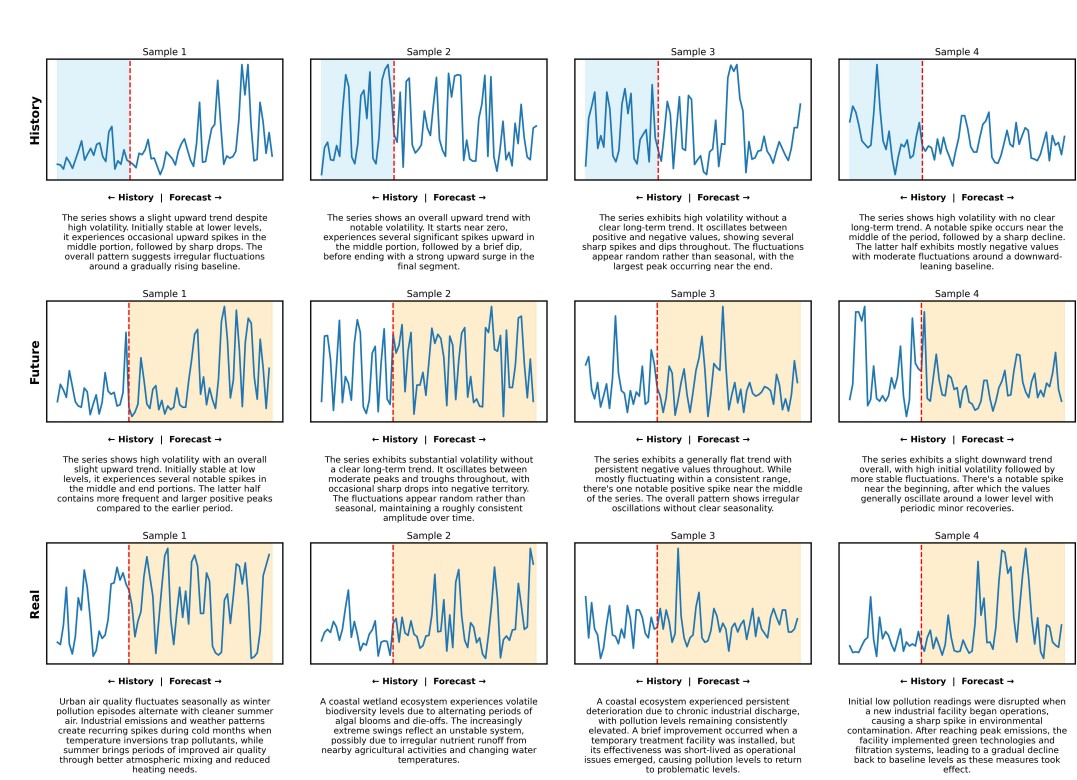

Figure 20: Synthetic text for Environment dataset.

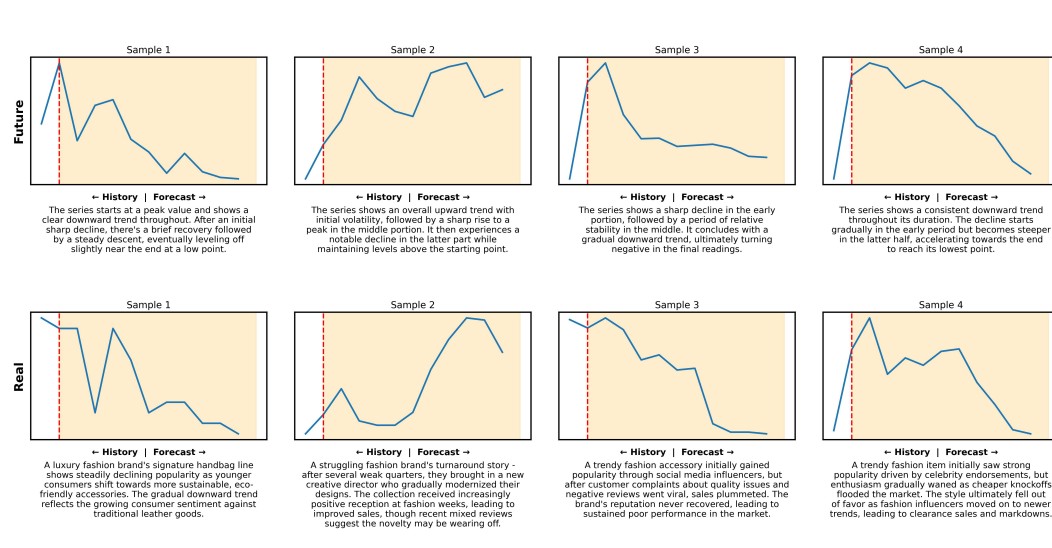

Figure 21: Synthetic text for Fashion dataset.

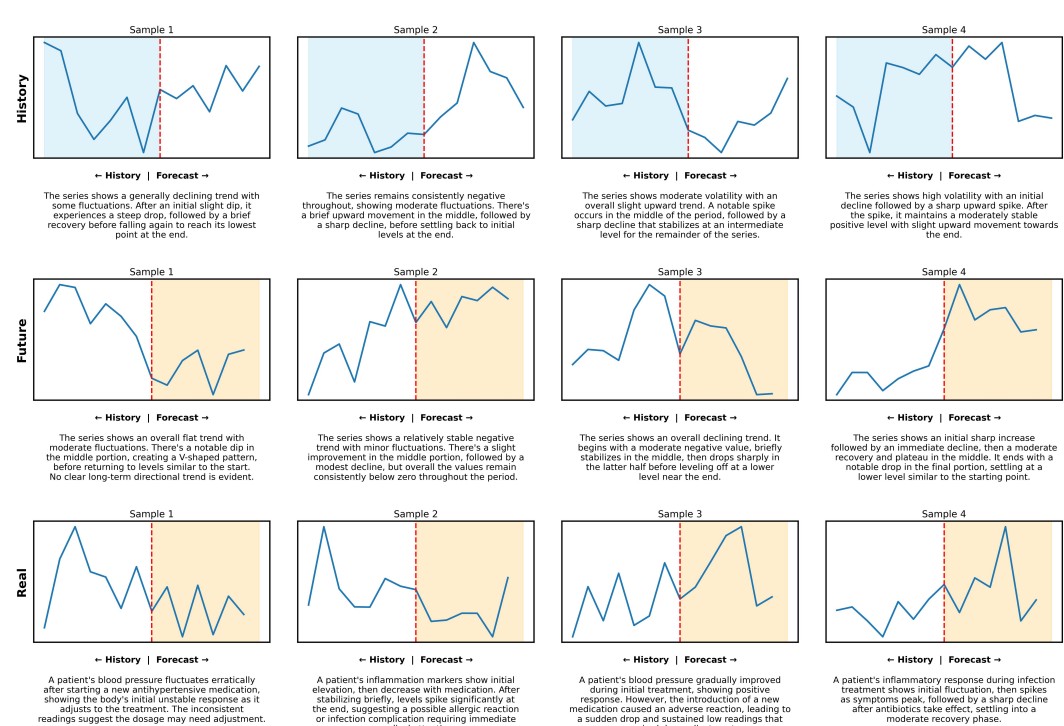

Figure 22: Synthetic text for Medical dataset.

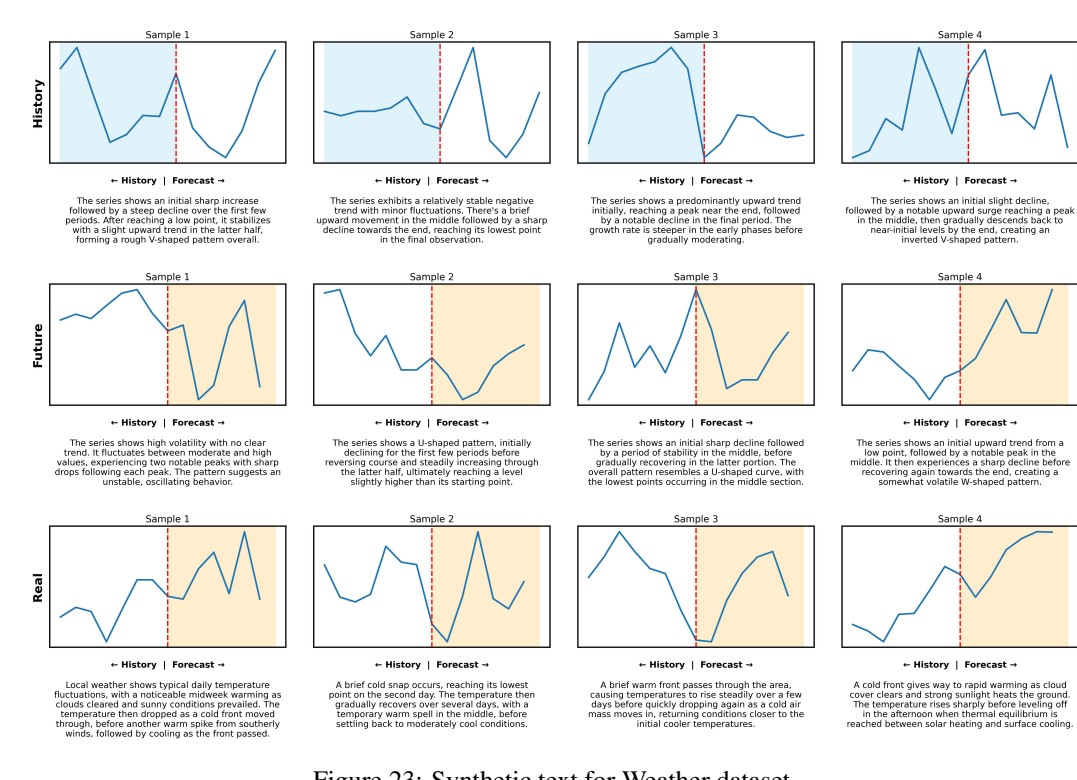

Figure 23: Synthetic text for Weather dataset.

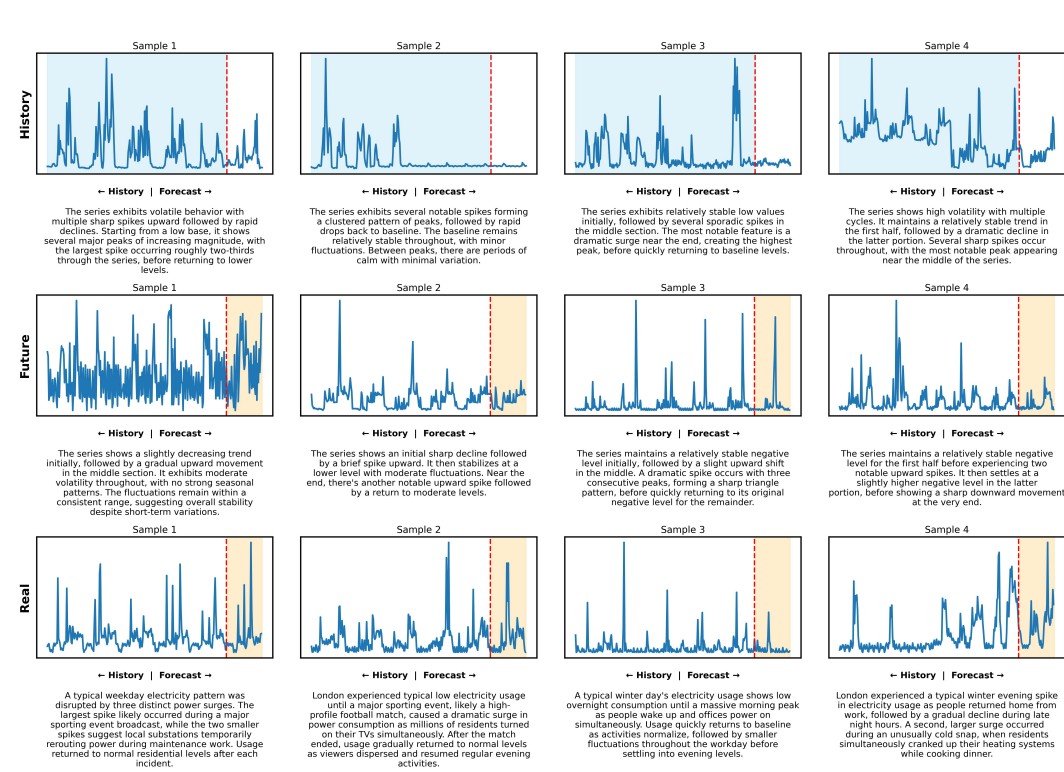

Figure 24: Synthetic text for LEU dataset.

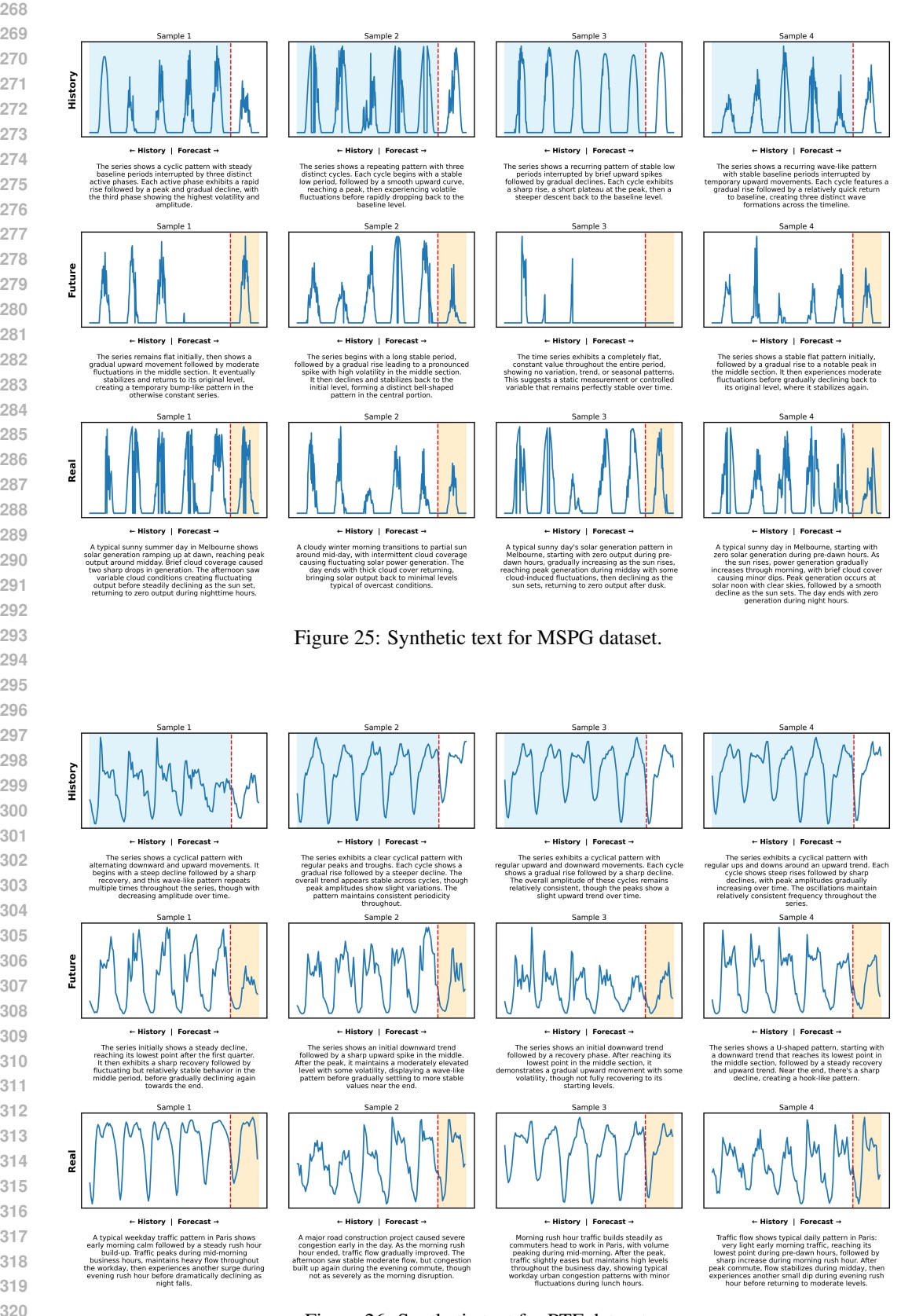

Figure 25: Synthetic text for MSPG dataset.

Figure 26: Synthetic text for PTF dataset.

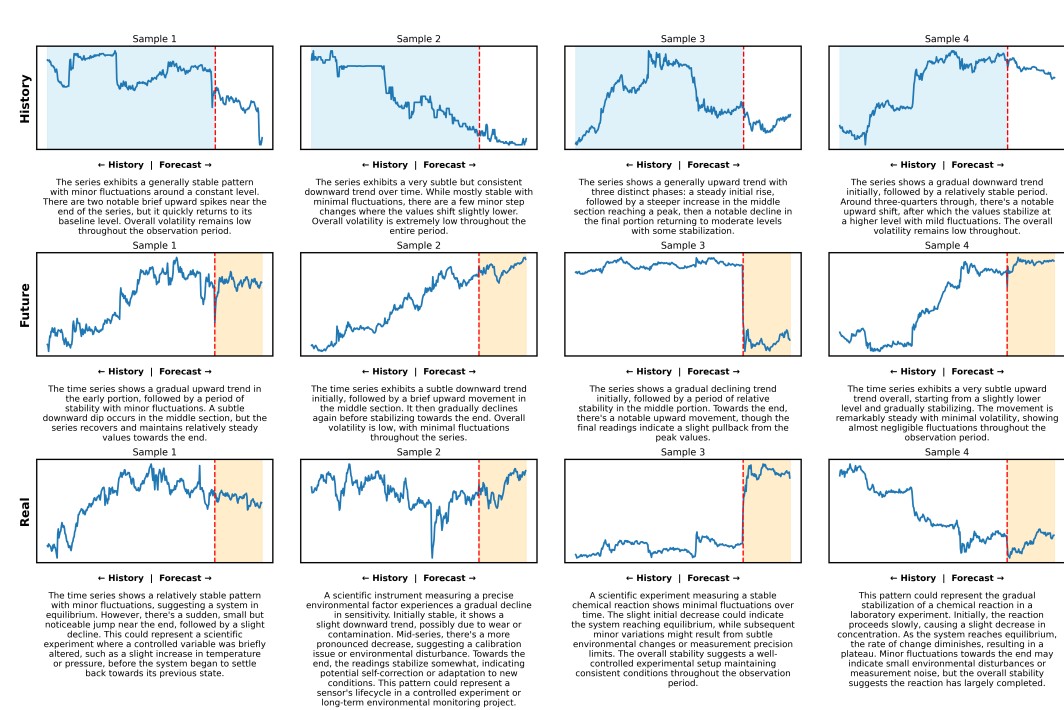

Figure 27: Synthetic text for MTFinance dataset.

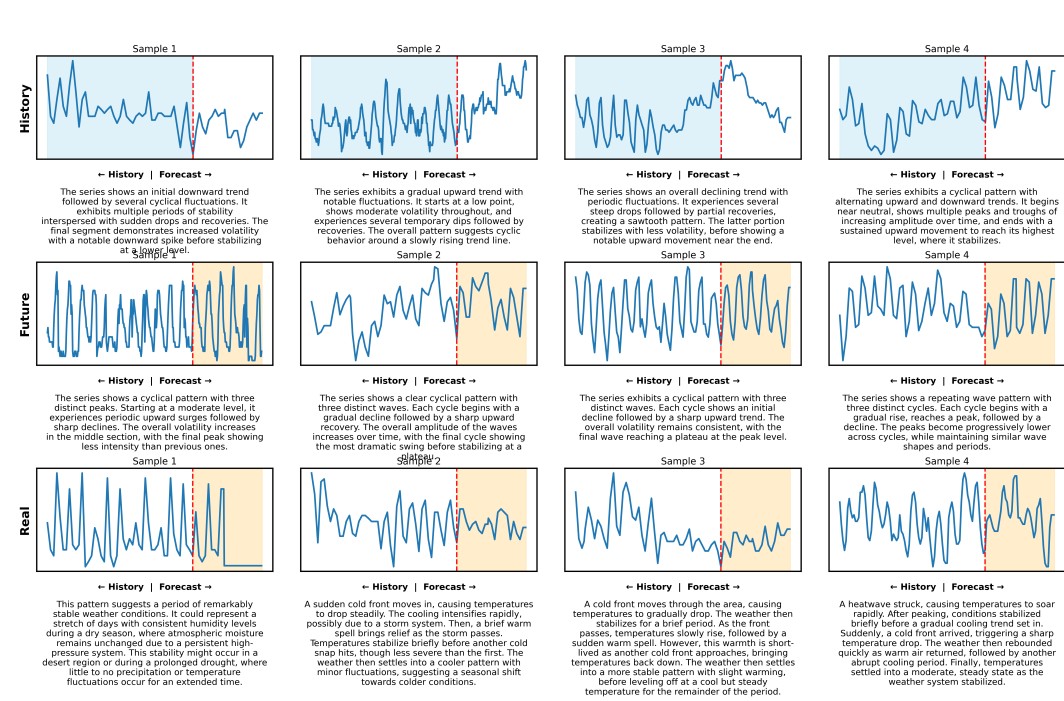

Figure 28: Synthetic text for MTWeather dataset.

Table 22: Domain hints provided to the LLM based on filename keywords.

| Filename Keyword | Domain Hint |
|---|---|
| agri | This dataset comes from the agriculture domain and may track crop yields, irrigation usage, or farm production activity. |
| climate | This dataset reflects climate variables such as temperature, precipitation, or CO2 levels over time. |
| economy | This dataset tracks economic indicators such as GDP, inflation, unemployment, or financial market performance. |
| energy | This dataset belongs to the energy sector and may measure electricity demand, power generation, or gas usage. |
| environment | This dataset relates to environmental monitoring, such as pollution levels, ecosystem health, or sustainability metrics. |
| health | This dataset comes from the healthcare domain and may track patient vitals, disease spread, or hospital resource use. |
| socialgood | This dataset is related to social good applications, potentially measuring outcomes in education, inequality, public health, or crisis response. |
| traffic | This dataset captures traffic flow, congestion, or transportation usage in a road or urban network. |
| fashion | This dataset is from the fashion or retail sector and may track product demand, seasonal sales trends, or consumer behavior. |
| weather | This dataset records weather conditions such as temperature, humidity, wind speed, or rainfall. |
| medical | This dataset comes from the medical domain and tracks clinical events, treatment responses, or patient monitoring data. |
| ptf | This dataset comes from Paris Traffic Flow |
| mspg | This dataset is from Melbourne Solar Power Generation. |
| leu | This dataset London Electricity Usage |
| (other) | This dataset belongs to a scientific or application-specific domain. Use best judgment to infer a plausible real-world context. |

## H   LIMITATIONS AND FUTURE WORK

Our study focuses on text and time series. However, real-world forecasting tasks may involve other modalities such as product images in retail forecasting. These additional modalities are not considered in this work, and extending our analysis to broader multimodal settings remains an important direction for future research. Moreover, although our benchmark spans 16 diverse datasets, they may not fully capture the breadth of challenges encountered in real-world applications. We leave the exploration of MMTS methods at greater scale and across a wider array of datasets as future work.

## I   THE USE OF LARGE LANGUAGE MODELS (LLMS)

LLMs are used to polish the language of the paper. Specifically, they help with improving grammar, clarity, and readability of sentences. No LLMs were used for research ideation or method design.

## J   MMTS WITH IMAGE MODALITY

We focus on the dual-modality case of time series and external text, which is the most common form of MMTS research domain. For the Fashion e-commerce forecasting dataset in Table 1, we ran new experiments incorporating product images in the dataset. For aligning-based models, we use the CLIP model to embed images and text, and use Chronos to extract time series features: Aligning (TS+image) and Aligning (TS+text+image). For prompting-based models, we use Claude 3.7 to process images: Prompting (TS+image) and Prompting (TS+text+image). Our conclusion remains consistent that adding the image information does not outperform the TS+text variant in Table 1. The TS+image and TS+text+image models are within noise of TS+text and sometimes slightly worse. Qualitatively, the images provide style/category redundancy already captured by textual tags, adding little complementary signal, which is consistent with our findings in Section 4.6. We show that our central claim holds for other modalities that the benefit of an additional modality is conditional on its complementarity and not just its presence.

Table 23: Performance on the Fashion dataset after adding the image modality.

| Metrics | Aligning | | Prompting | |
|---------|------------|-------------------|------------|-------------------|
| | Image + TS | Image + Text + TS | Image + TS | Image + Text + TS |
| MSE | 0.526 | 0.490 | 0.659 | 0.677 |
| MAE | 0.520 | 0.486 | 0.574 | 0.576 |

## K   ADDITIONAL PARADIGM BASED ON CODE GENERATION

Recently, there exist additional multimodal paradigms based on reinforcement learning and code generation Zhang et al. (2025); Tiomoko et al. (2025). However, these works are not designed for multimodal forecasting. TimeMaster Zhang et al. (2025) is for time series classification but not for forecasting. HITL Tiomoko et al. (2025) does not handle multimodal inputs for forecasting but converts human instructions into post-processing code to correct an existing forecast. We adapted the reinforcement learning and code generation ideas from TimeMaster and HITL to our multimodal forecasting setting. We revised TimeMaster for code generation using the Qwen3-32B model for multimodal forecasting. We prompted Qwen3-32B model to take in time series and context, and to generate code with the most appropriate feature preprocessing technique and modeling paradigm considering unimodal, aligning, and prompting-based paradigms. Results are worse than aligning-based methods as shown in Table 24. This supports our conclusion that directly leveraging LLMs for numerical forecasting (through prompting or code generation) still has a challenging gap.

Table 24: Performance of code generation paradigm.

| Dataset | MSE | MAE |
|---|---|---|
| Agriculture | 7.469 | 2.161 |
| Climate | 1.291 | 0.888 |
| Economy | 6.328 | 2.213 |
| Energy | 1.002 | 0.738 |
| Environment | 2.932 | 0.780 |
| Health | 2.403 | 0.959 |
| Socialgood | 2.113 | 1.017 |
| Traffic | 1.774 | 1.179 |
| Fashion | 0.541 | 0.514 |
| Weather | 0.609 | 0.644 |
| Medical | 2.239 | 1.088 |
| PTF | 5.052 | 0.816 |
| MSPG | 1.473 | 0.555 |
| LEU | 1.106 | 0.566 |
| MTFinance | 0.793 | 0.374 |
| MTWeather | 0.900 | 0.727 |

