# OpenReview forum: "When Does Multimodality Lead to Better Time Series Forecasting?"
_ICLR.cc/2026/Conference — Submitted to ICLR 2026_

### Official Review · Reviewer_GVvr · 2025-10-25

**Soundness:** 2
**Presentation:** 2
**Contribution:** 2
**Rating:** 2
**Confidence:** 5

**Summary:**

This paper investigates whether and under what conditions the textual modality benefits time series forecasting. Aligning-based methods and prompting-based methods are summarized for experiments. Experimental results reveal that the benefits of textual modality are highly condition-dependent for time series forecasting.

**Strengths:**

1. This paper explores the contribution of textual modality and large language models (LLMs) to time series forecasting, with particular attention to the effect of model size.

2. The experimental evaluation is extensive and well-organized.

3. The paper covers a wide range of domains, enhancing the comprehensiveness of the analysis.

**Weaknesses:**

1. The existing work CM2TS [1] has already investigated similar questions regarding cross-modality modeling for time series. However, this paper does not provide a proper citation or discussion to clarify how its contributions differ from or extend CM2TS.

2. Only text/language is involved as an external modality. Thus, this study is a dual-modality or cross-modality analysis, rather than a multimodality analysis. Please ensure the authenticity.

3. Some experimental results (e.g., Sections 4.2 and 4.5) are interesting but appear to depend heavily on hyperparameter configurations of methods. As such, the observed improvements may correspond to local optima. It would be better to combine the theoretical proof and the absolutely global optimal experimental results to verify your assumption.

4. The code and implementation details are not available, which limits reproducibility and independent validation of the findings.

[1] Towards Cross-Modality Modeling for Time Series Analytics: A Survey in the LLM Era, IJCAI 2025.

**Questions:**

1. It is difficult to understand Table 1. How are PatchTST, DLinear, and Chronos aligned with LLMs? Which alignment strategies were applied in each case?

2. For each LLM-based method, have you fine-tuned the best hyperparameters individually, or did you use a unified setting across models? Please clarify the procedure.

3. The paper discusses the benefit of multimodality, but it seems that CM2TS [1] has already explored a similar topic. Could the authors clarify how this work differs from or extends the contributions of CM2TS?

[1] Towards Cross-Modality Modeling for Time Series Analytics: A Survey in the LLM Era, IJCAI 2025.

---

> ### Author Response · Authors · 2025-11-20
>
> We sincerely thank the reviewer for the feedback. We appreciate that the reviewer finds (1) our evaluation extensive and well-organized; (2) our analysis comprehensive covering a wide range of domains. We address the reviewer’s concerns as follows.
>
> > Compared to CM2TS.
>
> We have added reference of CM2TS in the updated introduction. We would like to clarify that compared to CM2TS, our paper **is not a survey paper** but a rigorous study that helps **understand under what conditions** does the inclusion of additional text helps, which **brings new contributions** to the time series community. We empirically answer this question from both modeling and data perspectives, and also provide **controlled synthetic experiments** to quantitatively study the effect of complementarity in text information, providing **insights that are generalizable across datasets**.
>
> > Other modalities.
>
> We use "multimodal" as it is the common, accepted term in the literature for this problem. To improve precision, we have added a note clarifying that our focus is on the dual-modality case of (TS + external text), which is the most common form of MMTS research domain.
>
> In addition, for the fashion e-commerce forecasting dataset in Table 1, we ran new experiments incorporating product images in the dataset. For aligning-based models, we use the CLIP model to embed images and text, and use Chronos to extract time series features: **Aligning (TS+image)** and **Aligning (TS+text+image)**. For prompting-based models, we use Claude 3.7 to process images: **Prompting (TS+image)** and **Prompting (TS+text+image)**. Our conclusion remains consistent that adding the image information does not outperform the TS+text variant. The TS+image and TS+text+image models are within noise of TS+text and sometimes slightly worse. Qualitatively, the images provide style/category redundancy already captured by textual tags, adding little complementary signal, which is consistent with our findings in Section 4.6. We show that our central claim holds for other modalities that the benefit of an additional modality is conditional on its complementarity and not just its presence.
>
> |Metric   |Aligning (TS + image)|Aligning (TS + text + image)|Prompting (TS + image) |Prompting (TS + text + image)|
> |---------|---------------------|----------------------------|-----------------------|-----------------------------|
> |MSE      | 0.526               |0.490                       | 0.659                 | 0.677                       |
> |MAE      | 0.520               |0.486                       | 0.574                 | 0.576                       |
>
> > Hyper-parameter optimization.
>
> We conducted hyper-parameter optimizations as detailed in **Appendix B and C**. For LLM-based methods, the major hyper-parameter is the temperature, which is optimized following CiK using a unified setting. For aligning-based method, we tuned e.g., the ratio of combining text embeddings with time series embeddings. In Section 4.4, we also provided detailed analysis on different aligning strategies. Our paper has provided rigorous empirical studies on when text helps multimodal forecasting. "Theoretical proof" for the "absolutely global optimal" for large-scale deep learning benchmark is outside the scope of an empirical paper, which we leave as future work.
>
> > Code and implementation details.
>
> We have provided extensive implementation details in **Appendix B and C** to ensure reproducibility. We have also added a dedicated **"Reproducibility Statement" section** in **Lines 486 - 492**. We are currently going through legal review of our code release, and we will make the code public upon acceptance of the paper.
>
> > How are PatchTST, DLinear and Chronos aligned with LLMs.
>
> We detailed the aligning strategies in **Appendix B**. In **Section 4.4**, we also analyzed different aligning strategies and found that “addition, average, residual projector, late fusion, and efficient fine-tuning” perform better on average, which is used as the default aligning strategy in Table 1.

---

> > ### Author Response · Authors · 2025-11-27
> >
> > Dear Reviewer GVvr
> >
> > We sincerely appreciate the reviewer’s time and thoughtful assessment of our work. We have made substantial improvements to the paper based on your feedback and have provided detailed explanations in our rebuttal. We hope that our clarifications and additional experiments fully address your concerns. Please let us know if any questions remain. Thank you once again for your time and effort.
> >
> > Best regards,
> > Authors

---

### Official Review · Reviewer_JcvS · 2025-10-31

**Soundness:** 2
**Presentation:** 3
**Contribution:** 2
**Rating:** 2
**Confidence:** 4

**Summary:**

In this work, the authors evaluate various existing models for multimodal time series (MMTS) forecasting across 16 datasets under two paradigms: alignment-based and prompt-based methods. The paper provides detailed analyses that can inform the broader understanding of this research field. However, the most critical aspect—namely, the definition of “multimodal time series” -- is not clearly explained.

**Strengths:**

- The authors evaluate multiple existing methods using two pipelines (alignment-based and prompt-based) across 16 diverse datasets.
- The paper provides a comprehensive analysis of different modeling strategies and offers detailed experimental results and insights.

**Weaknesses:**

### 1. **The Definition of “Multimodal” Time Series**

I am very concerned about the formulation of **multimodal** time series in this paper.

In Lines 034–044, the authors categorize six “MMTS” methods into two types: (I) alignment-based and (II) prompt-based.

However, I believe that **multimodal learning inherently implies semantic alignment** between modalities. For instance, I think it difficult to perceive any semantic alignment between numerical time series data and a textual statement such as *“Tomorrow there will be a meeting between the US and Canada.”* These two sources do not describe the same underlying content; rather, they convey fundamentally different types of information. Hence, such data and methods are better described as **multi-source** or **multi-factor** forecasting approaches, rather than traditional multimodal learning.

**Speech–text** is a good example to illustrate what true semantic alignment means. The alignment between modalities presupposes that they represent the *same underlying content* (e.g., the same utterance in two modalities) rather than *different kinds of information*. In contrast, most time series language models, e.g., [4],  do not share this semantic correspondence.

For multimodal time series (MMTS), recent works [1, 2, 3] have explored more semantically grounded formulations by transforming raw time series into frequency, visual, or textual representations—e.g., spectrograms or pattern images—and learning alignment in these shared spaces. These directions capture the essence of multimodality much better.

If the authors intend to redefine or extend the concept of MMTS, I strongly suggest that they **explicitly discuss what constitutes a multimodal time series and what does not**. As it stands, I disagree with the paper’s implicit definition of “multimodal time series,” which appears more closely related to multi-source or multi-factor data integration rather than genuine multimodal learning.


[1] Time-VLM: Exploring Multimodal Vision-Language Models for Augmented Time Series Forecasting

[2] Teaching Time Series to See and Speak: Forecasting with Aligned Visual and Textual Perspectives

[3] GEM: Empowering MLLM for Grounded ECG Understanding with Time Series and Images

[4] One Fits All: Power General Time Series Analysis by Pretrained LM

### 2. **Benchmark**

Considering the conceptual gap between conventional multimodal learning and the “MMTS” formulation in this paper, I believe it is necessary for the authors to provide a more comprehensive clarification of the semantic relationship between time series and textual modalities in the chosen datasets. Moreover, recent studies have demonstrated that vision-language models (VLMs) can effectively comprehend and reason over time-series data, highlighting the importance of discussing these conceptual distinctions in greater depth.

**Questions:**

For my question, please refer to the “Weaknesses” section.

Additionally, although the authors provide extensive experimental details in Appendices A, B, and C, following ICLR’s reproducibility guidelines, it is recommended that the authors include a dedicated “Reproducibility Statement” section before the References.

---

> ### Author Response · Authors · 2025-11-20
>
> We sincerely thank the reviewer for the feedback. We appreciate that the reviewer recognizes that we provided comprehensive and detailed analysis that inform broader understanding of the research field. We address the reviewer’s concerns as follows.
>
> > Definition of multimodal time series.
>
> 1. We would like to clarify that our definition for this work follows the **established convention** within the time series community, where text provides crucial **contextual or event-based information** that is external from time series data itself **[1-7]**. We explicitly define our scope in the Introduction (**Lines 034-036**) as combining time series data with **auxiliary textual sources** (e.g., clinical notes, product descriptions). We also refer to other well established studies with this definition **[1-7]**. For example:
>
>     * Time-MMD [1]: mentions “exogenous or auxiliary textual signals”.
>     * Context is Key [3]: focuses on “produce statistical forecasts by incorporating relevant side information (i.e., context)”.
>     * LLMP [6]: mentions “guided by natural language text which describes a user’s prior knowledge”.
>     * Even the Time-VLM paper that the reviewer referenced defines their scope as “augmenting time series forecasting with additional modalities, such as text and images, which provide complementary information that can enhance predictive accuracy”. They use domain knowledge as text modality that represents external information from the numerical time series (e.g., Electricity consumption typically peaks at noon, drop at night).
>
> 2. Describing time series directly as text suggested by the reviewer is also covered by our analysis in **Section 4.6**. We use LLMs to generate text that purely describe the history of time series (**History Desc in Figure 11**). We found this performs worse than text that contains more complementary and predictive information as shown in Figure 11.
>
> 3. We also added an experiment **converting time series into images** (TS-as-image) and used a vision language model Claude 3.7 to forecast in **Appendix E.5**. We compared three scenarios: **TS-as-image+text+TS**, **text+TS** and **TS-as-image+TS**, which perform comparably with each other. These results are consistent with our analysis, which also point to the importance of complementary, future-predictive information.
>
> [1] Time-MMD: Multi-Domain Multimodal Dataset for Time Series Analysis
>
> [2] ChatTime: A Unified Multimodal Time Series Foundation Model Bridging Numerical and Textual Data
>
> [3] Context is Key: A Benchmark for Forecasting with Essential Textual Information
>
> [4] Context Matters: Leveraging Contextual Features for Time Series Forecasting
>
> [5] Multi-Modal Forecaster: Jointly Predicting Time Series and Textual Data
>
> [6] LLM Processes: Numerical Predictive Distributions Conditioned on Natural Language
>
> [7] GPT4MTS: Prompt-based Large Language Model for Multimodal Time-series Forecasting
>
> We address the remaining questions in the following comment.

---

> ### Author Response · Authors · 2025-11-20
>
> > Vision-language model.
>
> We additionally converted time series into images (TS-as-image) and compared with a **vision-language model** Claude 3.7 (results updated in **Appendix E.5**). We compared three scenarios: **TS-as-image+text+TS**, **text+TS** and **TS-as-image+TS**, which perform comparably with each other. These results are consistent with our analysis on the text modality, which also point to the importance of complementary, future-predictive information. Our conclusion also remains that numerical reasoning for forecasting remains a significant challenge for current LLMs, regardless of size or architecture.
>
> |             | VLM        (TS-as-image + Text + TS) |         | VLM        (Text + TS) |         | VLM        (TS-as-image + TS) |         |
> | ----------- | ------------------------------ | ------- | ---------------------- | ------- | ----------------------- | ------- |
> |             | MAE                            | MSE     | MAE                    | MSE     | MAE                     | MSE     |
> | Agriculture | 0.236                          | 0.122   | 0.242                  | 0.132   | 0.246                   | 0.133   |
> | Climate     | 0.956                          | 1.464   | 0.984                  | 1.492   | 0.967                   | 1.487   |
> | Economy     | 0.164                          | 0.043   | 0.132                  | 0.027   | 0.158                   | 0.039   |
> | Energy      | 0.228                          | 0.118   | 0.244                  | 0.133   | 0.252                   | 0.156   |
> | Environment | 0.581                          | 0.646   | 0.590                  | 0.694   | 0.584                   | 0.651   |
> | Health      | 1.180                          | 5.276   | 1.140                  | 2.966   | 1.333                   | 5.949   |
> | Socialgood  | 0.450                          | 0.897   | 0.419                  | 0.734   | 0.461                   | 0.937   |
> | Traffic     | 0.382                          | 0.318   | 0.238                  | 0.168   | 0.408                   | 0.358   |
> | Fashion     | 0.502                          | 0.592   | 0.520                  | 0.565   | 0.505                   | 0.609   |
> | Weather     | 0.268                          | 0.258   | 0.276                  | 0.271   | 0.264                   | 0.249   |
> | Medical     | 0.812                          | 0.523   | 0.858                  | 0.602   | 0.794                   | 0.511   |
> | PTF         | 0.339                          | 0.304   | 0.307                  | 0.247   | 0.440                   | 0.464   |
> | MSPG        | 0.321                          | 0.570   | 0.291                  | 0.466   | 0.396                   | 0.828   |
> | LEU         | 0.489                          | 0.827  | 0.429                  | 0.755   | 0.436                   | 0.766   |
> | MTFinance   | 0.013                          | 0.002  | 0.012                  | 0.002   | 0.013                   | 0.002   |
> | MTWeather   | 0.482                          | 0.432  | 0.496                  | 0.441   | 0.507                   | 0.449   |
> | Average     | 0.462                        | 0.649 | 0.448                | 0.605 | 0.485                 | 0.849 |
>
> > Reproducibility section.
>
> We have provided extensive implementation details in the appendix to ensure reproducibility. We also mentioned in the **last line of conclusion (Lines 484 - 485)** that “Reproducibility details are discussed in Appendix B and Appendix C.” Furthermore, we have **added a dedicated "Reproducibility Statement" section on Lines 486 - 492**.

---

> ### Comment · Reviewer_JcvS · 2025-11-21
> **Re: Clarification on Definitions, The "Semantic Gap"  and Scientific Precision**
>
> Thanks for the additional experiments and clarifications. However, the authors did not directly address my primary concern regarding the definition and mechanism of "multimodality."
>
> The response mainly reiterates existing conventions without engaging with the conceptual distinction I raised between multimodal/multi-factor/multi-source augmentation, multimodal semantic alignment, and related work [8-10].
>
> However, these concerns are central to the paper's stated research question, as the title: ***"When Does Multimodality Lead to Better Time Series Forecasting?"***
>
> To scientifically answer **"When"** a method works, one must first rigorously define **"What"** the mechanism is.
>
> I will prepare a more detailed revision later, but I wanted to share these main points first.
>
> [8] Time-VLM: Exploring Multimodal Vision-Language Models for Augmented Time Series Forecasting
>
> [9] Teaching Time Series to See and Speak: Forecasting with Aligned Visual and Textual Perspectives
>
> [10] GEM: Empowering MLLM for Grounded ECG Understanding with Time Series and Images

---

> > ### Author Response · Authors · 2025-11-21
> >
> > Thank the reviewer for the follow up. We would like to clarify that the semantic alignment across modality pairs is not binary but rather spans a continuous spectrum.
> >
> > * Speech-Text shows relatively high alignment, but even here, the modalities contain complementary information. For example, text fails to capture prosody, emotion or vocal texture.
> > * Image-Text falls in the middle of the spectrum, where visual details often lack direct syntactic equivalents in text, and textual descriptions cannot fully specify the richness of visual content.
> > * TS-Text lies on the lower end of the spectrum as the semantic relationship is sporadic, dynamic and probabilistic. For example, positive earnings news may correspond only to a probabilistic change within a local future window of the stock time series.
> >
> > We clarify that our work fundamentally investigates the utility of auxiliary external text as mentioned in Lines 034-036. We agree that this scope fits broadly within a multi-source or multi-factor taxonomy. We are happy to clarify this and revise the title or text if needed.

---

> > > ### Author Response · Authors · 2025-11-27
> > >
> > > Dear Reviewer JcvS
> > >
> > > We sincerely appreciate the reviewer’s time and thoughtful assessment of our work. We have made substantial improvements to the paper based on your feedback and have provided detailed explanations in our rebuttal. We hope that our clarifications and additional experiments fully address your concerns. Please let us know if any questions remain. Thank you once again for your time and effort.
> > >
> > > Best regards,
> > > Authors

---

### Official Review · Reviewer_v77y · 2025-11-01

**Soundness:** 3
**Presentation:** 3
**Contribution:** 3
**Rating:** 6
**Confidence:** 4

**Summary:**

This work systematically addresses the fundamental question of when and why multimodality improves forecasting, from both model and data perspectives. By evaluating 16 benchmarks and two key paradigms (alignment-based and prompting-based), the study offers essential insights into choosing encoders and fusion strategies for building powerful multimodal forecasting models.

**Strengths:**

1. The research is underpinned by a clearly defined and compelling motivation.

2.  The work itself is pioneering and addresses a problem of considerable importance.

3. The article is exceptionally well-written, and the experimental section is systematically conducted, with results presented in a clear and convincing manner.

**Weaknesses:**

Please refer to the **Questions**.

**Questions:**

The paper validates through synthetic and real-world datasets: MMTS is effective only when the text provides complementary predictive signals not contained in the time series. However, this conclusion is based on static evaluation scenarios (i.e., the "complementarity" between text and time series in experiments is fixed, such as whether the text contains trend shift information in synthetic data being a predefined condition). In real-world scenarios, the complementarity of text often changes dynamically (for example, in economic time series forecasting, a piece of news may contain complementary information before a policy is released but becomes redundant after the policy is implemented; in medical monitoring, the complementarity of clinical notes dynamically evolves with the patient's condition stability). Existing MMTS models all adopt fixed fusion strategies (such as fixed late fusion or early fusion) and are unable to determine in real time whether the text possesses complementarity and adjust the fusion intensity. Therefore, how can we design an MMTS model that can dynamically quantify the complementarity intensity between text and time series (e.g., based on metrics such as novelty in temporal patterns or semantic relevance of text) and adaptively switch fusion strategies (e.g., increasing text weight when complementarity is high, or reverting to unimodal mode when complementarity is low)? Can such a dynamic mechanism break through the performance ceiling of existing static fusion models in real-world scenarios (such as real-time economic forecasting or intensive care time series monitoring)?

---

> ### Author Response · Authors · 2025-11-20
>
> We sincerely thank the reviewer for the support and for recognizing our work as “pioneering” and “exceptionally well-written”. We are particularly grateful for the insightful question, which we believe perfectly frames the next step in this research area.
>
> > Quantify the complementarity and adaptively switch fusion strategies.
>
> Our paper provides the first large-scale static validation that "complementarity" is the key driver of MMTS performance (e.g., **Sec 4.6, Table 3, Figure 11**). We see our work as establishing the pre-condition and motivation for the very dynamic approach the reviewer describes.
>
> We explored various ways to quantify complementarity with results shown as below. We experimented on synthetic datasets where we know by construction that the unique text should have higher complementarity scores than redundant text. We tried using an **LLM as a judge (first table)** to score the uniqueness of the text data. Specifically, we used Claude 3.5 to assign a score of 0 or 1, 0 for unique text and 1 for redundant text, and computed the average score across each synthetic dataset. We also added two in-context learning (**ICL**) examples of paired time series and text for LLM as a judge (**second table**). However, these approaches did not yield a reliable signal across datasets that correlated with text complementarity or predictive performance gains. This difficulty shows that designing a dynamic approach that can effectively learn to measure and adapt to complementarity is a significant, complex, and new research challenge that is beyond the scope of our current analysis paper. Our findings directly motivate the need for an adaptive model, and we position the development of such a working solution as our direction for future work.
>
> |LLM as a Judge|Unique Text|Redundant Text|
> |--------------|-----------|--------------|
> |Trend      | 0.215     |0.315         |
> |Seasonality       | 0.900     |0.865         |
> |Spike      | 0.520     |0.615         |
>
> |LLM as a Judge (ICL)|Unique Text|Redundant Text|
> |--------------------|-----------|--------------|
> |Trend            | 0.470     |0.590         |
> |Seasonality             | 0.765     |0.905         |
> |Spike            | 0.550     |0.570         |

---

### Official Review · Reviewer_KtuZ · 2025-11-03

**Soundness:** 4
**Presentation:** 3
**Contribution:** 3
**Rating:** 4
**Confidence:** 3

**Summary:**

The paper "When Does Multimodality Lead to Better Time Series Forecasting?" conducts a systematic investigation into the effectiveness of multimodal time series (MMTS) forecasting by integrating textual information. The authors evaluate two dominant paradigms: alignment-based methods (fusing time series and text representations) and prompting-based methods (directly using LLMs for forecasting). Through a comprehensive benchmark spanning 16 datasets across 7 domains (e.g., health, economics), the study reveals that multimodal improvements are highly conditional and not universal. Key contributions include:

Demonstrating that MMTS methods do not consistently outperform unimodal baselines, challenging common assumptions.

Providing insights into how model capacity (e.g., text encoder size, time series model strength) and data characteristics (e.g., training data size, text complementarity) influence performance.

Offering data-agnostic guidelines via controlled experiments, such as synthetic data analyses, to generalize findings beyond specific benchmarks.

The paper emphasizes that multimodality is most beneficial when text provides complementary signals not captured by time series alone, and it encourages more cautious, data-driven approaches in future MMTS research.

**Strengths:**

Quality: The experimental design is thorough, covering 16 datasets, multiple model families (e.g., Chronos, BERT, LLMs), and diverse alignment strategies. The synthetic data approach is particularly strong for isolating key variables.

Clarity: The writing is accessible, with clear explanations of methods and results. Visualizations (e.g., scatter plots showing performance trends) effectively communicate complex findings.

Significance: The paper provides actionable guidelines for researchers and practitioners, potentially reducing wasted effort on ineffective multimodal integrations. Its focus on data characteristics beyond model architecture broadens the impact.

**Weaknesses:**

The study is limited to text and time series; excluding other modalities (e.g., images in retail forecasting) may reduce generalizability to broader multimodal settings.

While datasets are diverse, they may not capture all real-world challenges (e.g., ultra-long sequences or low-resource domains). Including more extreme cases could strengthen the conclusions.

The evaluation of prompting-based methods relies on current LLMs (e.g., GPT-4, Claude), which evolve rapidly; however, this is mitigated by testing multiple models and versions.


Some recent work addresses these problems by converting text into code or by involving human intervention to improve alignment (e.g., https://arxiv.org/abs/2505.15354, https://arxiv.org/pdf/2506.13705). We encourage future studies to demonstrate and benchmark such approaches as well.

**Questions:**

How might the inclusion of other modalities (e.g., images or audio) affect the conclusions? Could the guidelines be extended to multimodal settings beyond text?

Could the findings apply to streaming or online learning scenarios where data arrives incrementally?

Based on the results, are there specific domain invariants (e.g., healthcare vs. finance) where multimodality is consistently beneficial or ineffective?

Some recent work addresses these problems by converting text into code or by involving human intervention to improve alignment (e.g., https://arxiv.org/abs/2505.15354, https://arxiv.org/pdf/2506.13705). Please also include these kinds of methods—and any others that have been overlooked—to ensure a more complete and rigorous evaluation. I will raise additional points as further work and study are incorporated.

---

> ### Author Response · Authors · 2025-11-20
>
> We sincerely thank the reviewer for the feedback. We appreciate that the reviewer finds (1) our experiments thorough and synthetic data approach strong; (2) our paper significant with actionable guidelines, (3) our writing clear and accessible. We address the reviewer’s concerns as follows.
>
> > Other modalities.
>
> Our focus is on the most common text modality, and we have noted this in the Limitation section in **Appendix Section H**.
>
> For the fashion e-commerce forecasting dataset in Table 1, we ran new experiments incorporating product images in the dataset. For aligning-based models, we use the CLIP model to embed images and text, and use Chronos to extract time series features: **Aligning (TS+image)** and **Aligning (TS+text+image)**. For prompting-based models, we use Claude 3.7 to process images: **Prompting (TS+image)** and **Prompting (TS+text+image)**. Our conclusion remains consistent that adding the image information does not outperform the TS+text variant. The TS+image and TS+text+image models are within noise of TS+text and sometimes slightly worse. Qualitatively, the images provide style/category redundancy already captured by textual tags, adding little complementary signal, which is consistent with our findings in Section 4.6. We show that our central claim holds for other modalities that the benefit of an additional modality is conditional on its complementarity and not just its presence.
>
> |Metric   |Aligning (TS + image)|Aligning (TS + text + image)|Prompting (TS + image) |Prompting (TS + text + image)|
> |---------|---------------------|----------------------------|-----------------------|-----------------------------|
> |MSE      | 0.526               |0.490                       | 0.659                 | 0.677                       |
> |MAE      | 0.520               |0.486                       | 0.574                 | 0.576                       |
>
> > Additional paradigms such as code generation.
>
> We thank the reviewer for these very recent works and we have updated them in **Appendix K**. We found that these papers, while interesting, do not address the multimodal forecasting problem that we study. TimeMaster is for time series classification not for forecasting, and the human-in-the-loop paper does not handle multimodal inputs for forecasting but converts human instructions into post-processing code to correct an existing forecast.
>
> However, to be thorough, we adapted the code generation idea to our multimodal forecasting setting. We revised TimeMaster for code generation using the Qwen3-32B model for multimodal forecasting. We prompted Qwen3-32B model to take in time series and context, and to generate code with the most appropriate feature preprocessing technique and modeling paradigm considering unimodal, aligning, and prompting-based paradigms. Results are worse than current aligning-based methods as follows. This supports our conclusion that directly leveraging LLMs for numerical forecasting (through prompting or code generation) still has a challenging gap.
>
> |Dataset      | MSE    | MAE    |
> |-------------|--------|--------|
> |Agriculture  | 7.469  | 2.161  |
> |Climate      | 1.291  | 0.888  |
> |Economy      | 6.328  | 2.213  |
> |Energy       | 1.002  | 0.738  |
> |Environment  | 2.932  | 0.780  |
> |Health       | 2.403  | 0.959  |
> |Socialgood   | 2.113  | 1.017  |
> |Traffic      | 1.774  | 1.179  |
> |Fashion      | 0.541  | 0.514  |
> |Weather      | 0.609  | 0.644  |
> |Medical      | 2.239  | 1.088  |
> |PTF          | 5.052  | 0.816  |
> |MSPG         | 1.473  | 0.555  |
> |LEU          | 1.106  | 0.566  |
> |MTFinance    | 0.793  | 0.374  |
> |MTWeather    | 0.900  | 0.727  |
>
> > More diverse datasets in the extreme cases.
>
> We agree that capturing all challenges is difficult and we believe our 16-dataset benchmark provides significant diversity. We explicitly study **varying domains, varying lengths** (context length varying from 1 for fashion dataset to 480 for MSPG), and **varying data sizes** (training samples varying from 267 for economy to 7968 for LEU). The fashion dataset of short context length is a direct example of a low-resource case. We also studied the effect of context length (**Figure 9**) and data sample size (**Figure 8**), which also address the low-resource question.
>
> We additionally run experiments on an extra-long time series dataset PixelRec [1] and update the results in **Appendix D.4 and Table 15**. This dataset captures short video behavior with user interactions aggregated into daily view series. Each time series has context length of 365 and future length of 943. We use Chronos model as an example, and its unimodal and aligning-based counterparts perform comparably on this extremely-long dataset as well, consistent with our main findings.
>
> [1] MoTime: A Dataset Suite for Multimodal Time Series Forecasting
>
> |Metric|Unimodal|Aligning|
> |------|--------|--------|
> |MSE   |0.281   |0.280   |
> |MAE   |0.163   |0.162   |
>
> We address the remaining questions in the following comment.

---

> ### Author Response · Authors · 2025-11-20
>
> > Relying on current LLMs.
>
> Our goal was not to benchmark a specific LLM, but to understand the scaling properties. **Figure 3** shows a durable, generalizable insight: performance correlates with general reasoning (MMLU-Pro), but even the best LLMs (like Claude) still lag behind strong unimodal models (Chronos). This conclusion that numerical reasoning remains a gap is more important than the performance of any single model and will likely hold true for some time.
>
> > Streaming or online learning scenarios.
>
> Our work focuses on the standard batch forecasting setting. We believe our findings provide a critical foundation for future online learning work. For example, complementarity is still the key in online setting. An online model would need to dynamically assess if the new text is "redundant" or "unique" (as per our **Sec 4.6, Table 3**). A streaming model that naively fuses all text might be worse than a unimodal model, as it would be distracted by redundant data. Another example is that our finding in **Sec 4.4 (Table 2)** that "late fusion" is a robust default is also highly relevant for streaming, as it handles asynchronous data (e.g., TS every second, text every hour) more robustly. Late fusion isolates the pure TS encoder from the (potentially stale) text, preventing the "representation pollution" that early fusion would cause. This aligns with our finding that late fusion "minimizes disruption to temporal encoding." In summary, while a full investigation is a significant future direction, our paper provides the key principles that would be necessary.
>
> > Domain invariants.
>
> This is precisely what our data-centric analysis in **Section 4.6** addresses. We did not find that "domain" (e.g., finance) is the invariant. The invariant is the quality and complementarity of the text. **Figure 18 in Appendix** shows this clearly. On datasets like MSPG, LEU, and PTF, the "Original Text" (horizontal line) performs nearly as well as synthetic "LLM Event" or "LLM Desc" text, indicating the real text is highly informative. On other datasets, the original text is no better than descriptions of history, so performance gains are minimal. This data-agnostic principle ("complementarity matters, not domain") is a key takeaway.

---

> > ### Comment · Reviewer_KtuZ · 2025-11-21
> >
> > Overall, the authors have effectively addressed the majority of the concerns raised in the review. The key conclusion regarding the complementarity of text modalities—rather than mere presence—is well-supported and resonates with similar insights from some 2024 papers (e.g., works on multimodal fusion emphasizing conditional benefits), which this study further validates through empirical evidence.
> >
> > Looking ahead, I believe future directions for time series forecasting should focus on better generalization (e.g., online or incremental learning), incorporation of diverse modalities, and dynamic model adaptation to enhance robustness.
> >
> > In summary, this is a solid contribution to the field, and I hope the authors continue to explore these avenues in subsequent work.

---

> > > ### Author Response · Authors · 2025-11-21
> > >
> > > We sincerely thank the reviewer for their support and for recognizing this work as a solid contribution to the field. We fully agree with these future directions and will continue exploring them in our subsequent research.

---

### Author Response · Authors · 2025-11-19
**General Response**

We appreciate all reviewers for the feedback. We are glad that reviewers find our **experiments and analysis comprehensive** (**KtuZ, v77y, JcvS, GVvr**), our **writing clear** (**KtuZ, v77y**) and our **findings insightful** (**KtuZ, v77y, JcvS**). We have addressed reviewers’ concerns in separate responses, and the major ones are summarized as follows.

* We tested with **image modality** (TS+image and TS+text+image) which confirmed our core findings;
* We added experiments for another recent paradigm based on **code generation**, and found it performs worse than aligning-based paradigm;
* We added the PixelRec dataset for **ultra-long time series**, confirming consistent results on extreme lengths;
* We added experiments using LLM as a judge, trying to **quantify the correlations** between text and time series;
* We **clarified that our definition** of multimodal forecasting follows the **established convention** in the time series community;
* We added experiments using **vision language model** which confirmed our core findings.

---

### Author Response · Authors · 2025-12-02
**Summary of Rebuttal**

Dear AC, SAC, PC,

We understand that we are in a hard period, and the new ICLR policies have significantly increased your workload. To help minimize the time you need to spend, we have summarized each reviewer's feedback and our corresponding responses below.

We would like to highlight that reviewers find our **experiments and analysis comprehensive** (**KtuZ, v77y, JcvS, GVvr**), our **findings insightful** (**KtuZ, v77y, JcvS**), and our **writing clear** (**KtuZ, v77y**).

Reviewer **KtuZ** (initial rating 4 → **after rebuttal 6**) stated that "**this is a solid contribution to the field**" and "**the key conclusion is well-supported**". Their concerns were addressed during the rebuttal.

Reviewer **v77y** (**initial rating 6**) has not replied to our response, but their initial rating is positive, saying that our work is "**pioneering and addresses a problem of considerable importance**", our paper is "**exceptionally well-written**", and "the **experimental section** is **systematically conducted**, with results presented in a **clear and convincing** manner".

Reviewer **JcvS** (initial rating 2) mostly has concern about the conceptual definition of multimodality, arguing that different modalities should convey the same semantic content. We clarified that our scope of using auxiliary text **follows established definitions of multimodality in the time series community**, with many example references. Moreover, **modalities do not need to, and typically cannot, convey exactly the same information**. Instead, they provide different degrees of complementary signals, with each modality contributing aspects that the others cannot capture. We showed through controlled experiments in our paper (**Section 4.6**) that text with complementary information brings more benefits than text that simply describes the history time series without any complementary information. **The reviewer did not respond further, and we believe our clarification directly addresses the concern**.

Reviewer **GVvr** (initial rating 2) **has not replied to our response**. The reviewer raised questions about comparisons to a survey paper and extension to other modalities. In our rebuttal, we clarified that our paper is not a survey paper but a rigorous study that helps **understand under what conditions** does the inclusion of additional text helps, **which provides insights that are generalizable across datasets and brings new contributions to the time series community**. We also added an additional experiment using the vision modality, where we observed consistent patterns. Note that **Reviewer KtuZ initially had similar concerns, which they considered resolved after our rebuttal**.

We appreciate the useful feedback from all reviewers, and believe the rebuttal has further strengthened the paper. Thank you again for your time and consideration.

Best regards,

Authors

---

### Meta-Review · Area_Chair_eR6P · 2025-12-31

**Summary:**

This paper presents a large-scale empirical study on when and why textual information improves time series forecasting. The authors evaluate both alignment-based and prompting-based methods across 16 datasets spanning seven domains.

The reviewers generally appreciate the extensiveness of the experimental evaluation, the clarity of presentation, the insights derived from the analysis, and the practical guidelines provided.

On the negative side, the following concerns were raised:
- W1. The scope is relatively narrow, focusing only on the combination of time series and text.
- W2. Some important concepts, especially, (1) alignment between text and time series and (2) complementarity, which may vary depending on factors such as timing and the semantic content of the text, are not clearly defined.
- W3. The results may depend on specific parameter choices and particular versions of large language models.

Among these, I believe W3 is difficult to address in a fundamental manner and is not a limitation unique to this work.
Regarding W1, simply extending the study by converting time series into images would not necessarily address the concern, as such conversion may not introduce genuinely complementary information.
Regarding W2, I agree that the authors should be more careful with their terminology, to avoid potential misinterpretation of their findings.

With W1 and W2 remaining insufficiently addressed, I do not find the paper ready for acceptance in its current form.

**Reviewer Concerns:**

W1–W3 have not been fully addressed.

**Reviewer Scores:**

Reviewer KtuZ may revise the score upward, but I do not expect the paper to receive enough overall support for acceptance through discussion.

---

### Decision · Program_Chairs · 2026-01-26

Reject